# Non-Polar Gallium Nitride for Photodetection Applications: A Systematic Review

Omar Al-Zuhairi [1,2,*] , Ahmad Shuhaimi [2,*] , Nafarizal Nayan [3] , Adreen Azman [2], Anas Kamarudzaman [2], Omar Alobaidi [4] , Mustafa Ghanim [5] , Estabraq T. Abdullah [6] and Yong Zhu [7]

1   Department of Physics, Faculty of Science and Mathematics, Universiti Pendidikan Sultan Idris, Tanjung Malim 35900, Perak, Malaysia
2   Low Dimensional Materials Research Centre, Department of Physics, Faculty of Science, Universiti Malaya, Kuala Lumpur 50603, Malaysia; adreen@nitride.org.my (A.A.); anask52@gmail.com (A.K.)
3   Microelectronic and Nanotechnology Shamsuddin Research Centre (MiNT-SRC), Faculty of Electrical and Electronic Engineering, Universiti Tun Hussein Onn Malaysia, Batu Pahat 86400, Johor, Malaysia; nafa@uthm.edu.my
4   Department of Medical Instrumentation Engineering, Alsalam University College, Baghdad 10047, Iraq; omar.r.zaki@alsalam.edu.iq
5   Department of Medical Instrumentation Engineering, Ashur University College, Baghdad 10047, Iraq; mustafa.ghanim@au.edu.iq
6   Department of Physics, College of Sciences, University of Baghdad, Baghdad 10047, Iraq; estabraqtalib@sc.uobaghdad.edu.iq
7   Queensland Micro- and Nanotechnology Centre, Griffith University, 170 Kessels Road, Nathan, QLD 4111, Australia; y.zhu@griffith.edu.au
*   Correspondence: omarayad@fsmt.upsi.edu.my (O.A.-Z.); shuhaimi@um.edu.my (A.S.); Tel.: +60-10-271-4328 (O.A.-Z.); +60-19-971-1454 (A.S.)

**Abstract:** Ultraviolet photodetectors have been widely utilized in several applications, such as advanced communication, ozone sensing, air purification, flame detection, etc. Gallium nitride and its compound semiconductors have been promising candidates in photodetection applications. Unlike polar gallium nitride-based optoelectronics, non-polar gallium nitride-based optoelectronics have gained huge attention due to the piezoelectric and spontaneous polarization effect–induced quantum confined-stark effect being eliminated. In turn, non-polar gallium nitride-based photodetectors portray higher efficiency and faster response compared to the polar growth direction. To date, however, a systematic literature review of non-polar gallium nitride-based photodetectors has yet to be demonstrated. Hence, the objective of this systematic literature review is to critically analyze the data related to non-polar gallium nitride-based photodetectors. Based on the pool of literature, three categories are introduced, namely, growth and fabrication, electrical properties, and structural, morphological, and optical properties. In addition, bibliometric analysis, a precise open-source tool, was used to conduct a comprehensive science mapping analysis of non-polar gallium nitride-based photodetectors. Finally, challenges, motivations, and future opportunities of non-polar gallium nitride-based photodetectors are presented. The future opportunities of non-polar GaN-based photodetectors in terms of growth conditions, fabrication, and characterization are also presented. This systematic literature review can provide initial reading material for researchers and industries working on non-polar gallium nitride-based photodetectors.

**Keywords:** gallium nitride; non-polar; photodetector; systematic review; UV

## 1. Introduction

Over the past few decades, III-V semiconductor materials have shown promising outcomes for optoelectronic applications such as light-emitting diodes (LEDs), laser diodes (LDs), and photodetectors [1–6]. The demands for ultraviolet (UV) photo-sensing applications have markedly increased owing to the potential advantages of UV-sensing capabilities [1,7,8]. These capabilities could cover a wide range of applications, such as missile

detections, solar radiation detection, flame detection, and bio-photonics [9–13]. In addition, the use of UVC radiation detection that was obtained from germicidal illumination sources was reported to be employed in surface sterilizations from COVID-19 [1]. In recent years, several attempts were accomplished to enhance the efficiency of UV photodetectors by using wide and direct bandgap semiconductor materials such as aluminum gallium nitride (AlGaN)/aluminum indium gallium nitride (AlInGaN), aluminum nitride (AlN), gallium nitride (GaN) and zinc oxide (ZnO), and other organic compounds [12,14–18]; the range of the bandgaps of these materials is 0.7–6.4 eV. Thus, these materials are deemed suitable for the use of wide bandgap photodetectors [17,19,20]. However, the exciton-binding energy of these materials is 40–52 meV, resulting in a hindrance of the devices' performance [16,21,22]. In contrast, GaN has temperature stability and high radiation hardness with the smallest exciton-binding energy of 25 meV. In turn, GaN can be extremely versatile for UV photodetection since the separation of electron and hole in the presence of UV radiation is efficient, with fast transit time and high responsivity [17]. In addition to the aforementioned, unlike silicon (1.1 eV), GaN has a direct bandgap (3.43 eV) that could lead to a remarkable potential in UV photodetectors with high thermal stability and conductivity [1,9,23].

It is noteworthy that GaN-based UV photodetectors have been grown along the conventional c-plane direction owing to availability of the lattice-matched substrates and the intrinsic growth direction [24–28]. Despite the high crystal quality of these devices, they still suffer from the inherit spontaneous and piezoelectric polarization, leading to a reduction in GaN-based photodetector detection and collection of photo-generated carries [7,9,20,29]. On the other hand, using the non-polar direction for the growth of a GaN-based photodetector would promote intrinsically zero polarization effects [17,20]. Therefore, a non-polar GaN-based photodetector would exhibit a faster response with higher efficiency. Several groups have reported the use of the non-polar growth direction to produce GaN-based photodetectors along the non-polar direction as an alternative for c-orientated photodetectors [19,23,30–32]. However, it is noteworthy that non-polar GaN grown on sapphire or silicon (Si) substrates portrayed low crystal quality (far from device requirement) with a broad full width at half maximum (FWHM) of ∼0.5–0.6° as compared to the c-orientated GaN growth (less than 0.1°) [2,7,19,23,30,33–35]. This is mainly attributed to the anisotropic crystallographic mismatch owing to dissimilar lattice structures and mismatches in the coefficients of thermal expansion between non-polar GaN epitaxial layers and the substrates [19,23,31,36,37]. Consequently, high generated densities of defects and dislocations in the non-polar GaN epitaxial layer could impair the devices' efficiency, such as response of rise and fall time, compared to c-plane photodetectors [38–41]. On the other hand, substantial crystal qualities were achieved with the use of SiC and free-standing GaN bulk substrates [42,43]. However, substrates are expensive, and they will be less competitive for mass production. Therefore, there is a huge potential to achieve GaN-based UV photodetectors with superior performance along non-polar orientations.

Jabbar et al. reported a short review of the recent findings of polar GaN-based photodiodes in terms of device application and GaN characterization [44]. In addition, a short review of the Si, SiC, and diamond, as well as III-V-based UV photodetectors, was reported in terms of device applications and geometries [45]. However, a systematic literature review (SLR) article of non-polar GaN-based photodetectors in terms of growth and fabrication, electrical properties, and structural, morphological, and optical properties has yet to be investigated. It should be noted that SLR articles have been known for their significance in obtaining adequate understanding regarding interest in different fields [42,46]. Therefore, an SLR article is a crucial approach to facilitate the development of non-polar GaN-based photodetectors. In addition, there is a huge demand to highlight the challenges, motivations, and future opportunities of non-polar GaN-based photodetectors due to the eliminated polarization effects, which could lead to a faster response with higher efficiency.

In this paper, hence, an SLR of the recent approaches to grow non-polar GaN-based photodetectors from 2015 to 2021 is demonstrated. Discussions that revolve around the growth conditions and fabrications, structural properties, and electrical properties are

accomplished in detailed in relation to the published approaches of non-polar GaN-based photodetectors. In addition, discussions of the literature regarding challenges, motivations, and future opportunities are also presented. Finally, bibliometric analysis is presented to provide a transparent, systematic, and reproducible review process based on scientific activity, measurement of science, and scientists [47,48]. It is noteworthy that the bibliometric analysis would promote more objective, reliable, and structured analysis to the body of knowledge of non-polar GaN-based photodetectors. It can also provide an insight into the trends over time and themes researched, the most prolific institutions and scholars, the most relevant current and future trends, and an illustration of the "big picture" of extant research [49,50].

## 2. Systematic Literature Review Protocol

The presented manuscript is based on an SLR approach that has been recognized for its impact on acquiring adequate understanding regarding interest in different fields [46,51,52]. It is an organized approach that can be utilized to determine a definitive topic in the literature. Such an approach applies a scientific and systematic process that can be further utilized to evaluate, select, and crucially assess the related research articles to obtain the body of knowledge that were acquired form previous research publications. It can also provide an improved vision of the conventional approaches due to the incorporation of the literature into a more precise, transparent, and reproducible method. It should also be noted that SLRs have portrayed remarkably structured and analytical methods not only for research synthesis and the ability to facilitate several studies from various scientific disciplines, but also for postgraduate students who are pursuing a basis in their research efforts [51]. The process of the SLR consists of several stages: (i) the research area identification to determine the field of the study, (ii) the search procedure of keywords to create a query (set of the research keywords), (iii) the selection of search criteria to identify the criteria within the taxonomy, (iv) the extraction of the data process to obtain the data from the academic digital databases, and (v) the data synthesis to draw a conclusion of the study analysis.

### 2.1. Information Source

In this SLR, six academic digital databases were used to search, download, filter, extract, and draft the current SLR, as illustrated in Figure 1. The related academic digital databases were (i) ScienceDirect, (ii) Web of Science (WOS), (iii) Institute of Physics (IoP), (iv) American Institute of Physics (AIP), (v) American Chemical Society (ACS), and (vi) Wiley. The aforementioned academic digital databases can offer different scientific literature across all domains, which is deemed adequate to cover the latest and most reliable literature for non-polar GaN-based photodetectors. The extracted study from the above academic digital databases applies to revealing the role of non-polar GaN semiconductor materials in the development of photodetectors. The literature search was completely conducted on the six major databases within six years (2015–2021).

### 2.2. Source Strategy

The search process was conducted in two stages: the first stage was accomplished in March 2021, after the major highlights of the manuscript were identified, and the second stage was carried out in August 2021 to assure the recent literature was included. The search process was initially performed in the advanced search boxes of the aforementioned scientific databases. For the search process, the conjunctive and disjunctive models with Boolean operators (i.e., AND, OR) were employed in this study, whereby three groups of keywords (i.e., queries) were used in the search process, as illustrated in Figure 1. It should be noted that the Boolean operators were used here owing to their data source and access capabilities. Thus, the choice of Boolean operators was more practical in this SLR.

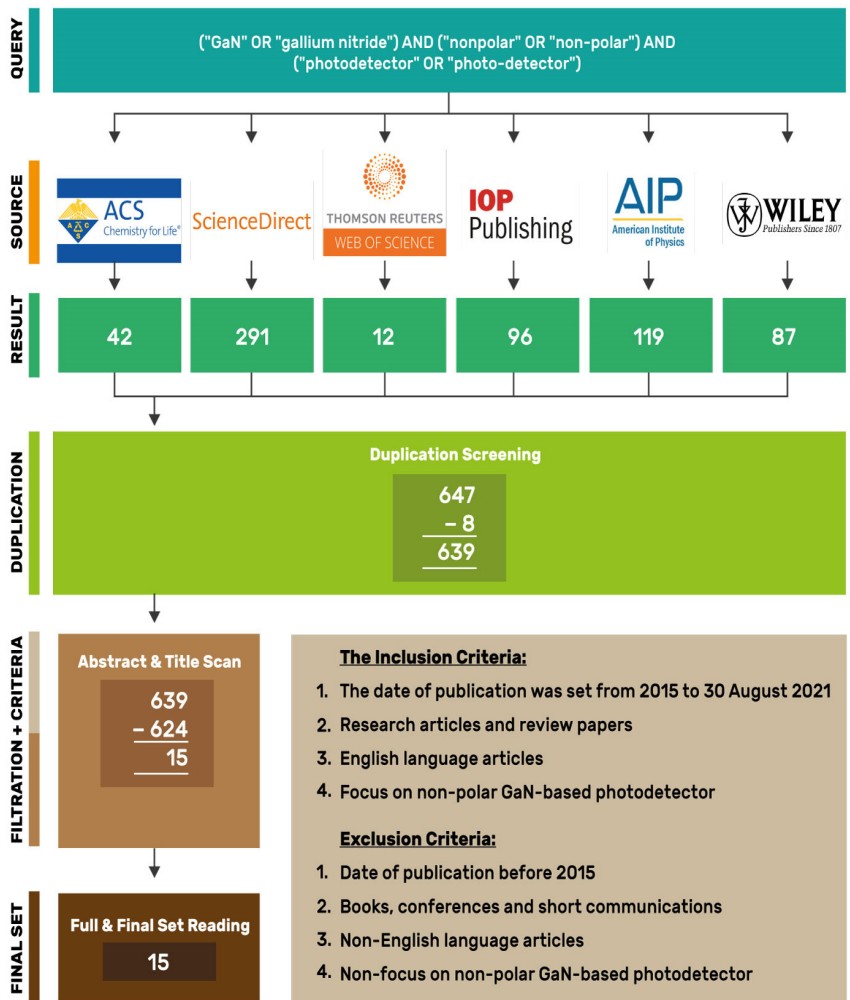

**Figure 1.** Flowchart of search query (("GaN" OR "gallium nitride") AND ("nonpolar" OR "nonpolar") AND ("photodetector" OR "photo-detector")) and inclusion criteria for the systematic review protocol.

### 2.3. Study Selection

The research process comprised three sub-processes, which were article collection, title and abstract scanning, and full-text reading. In the first process, the initial number of *n* = 647 was obtained from the above six academic digital databases, in which *n* = 8 duplicated articles were found and scanned over the aforementioned databases. In the second process, the articles were further examined by scanning their titles and abstracts, for which an investigation of the extracted articles was thoroughly performed to assure the inclusion or exclusion criteria were obtained. In the final stage, a full-text reading process was conducted to exclude the unrelated articles that did not meet the criteria of this review; *n* = 624 were identified as unrelated. A full-text reading process was carried out to extract the useful and valuable information (inclusion criteria) in relation to the final set of the articles for this review (*n* = 15). Several date extraction elements were collected during the process, as illustrated in Figure 1, which were included to provide insights to shape the latest vision of this review.

### 2.4. Data Extraction

Various points were analyzed via the collection and extraction of the data that were obtained from the articles, as shown in Figure 1. These attributes were the most crucial categories to gain the discussion points of this SLR, followed by the summary and description of each category. It should be noted these data extractions were selected for specific reasons:

They needed to be related to the title of the articles to lead to future referencing and/or the readers. In addition, the publication year and type, as well as the digital databases, were extracted to show the demographic statistics of the SLR to the readers. In turn, the publication progress from 2015 to 2021 is shown by presenting the number of studies of this topic. The publication type can provide a clear vision for the readers about the interest of the journals in relation to the aforementioned topic to acquire proper knowledge for potential future work. Furthermore, highlighting the challenges informs the reader and/or researchers about the problems in relation to non-polar GaN-based photodetectors that were encountered by previous research groups and provides a new direction for future research. Delineating the motivation is a way to describe the authors' demonstration process for future research groups to highlight the importance and the advantages of research on non-polar GaN-based photodetectors. Finally, the future opportunities in terms of growth conditions, fabrication, and characterization detail the significance of the attributes that should be investigated in future research.

### 2.5. Inclusion and Exclusion Criteria

It is noteworthy that several inclusion and exclusion criteria were employed while identifying the related articles in the process of the study selection, as illustrated in Figure 1. The publication date was set from 2015 to August 2021. In addition, all papers included within this review, such as research articles and review papers, were limited to the English language across all scientific databases. The inclusion criteria were concerned with papers that discussed non-polar GaN-based photodetectors. In contrast, according to the exclusion criteria, papers older than 2015 and not written in English were excluded. Finally, the papers that did not discuss non-polar GaN-based photodetectors were also excluded.

### 3. Taxonomy

The taxonomy that summarizes the search process results are presented in this section, as shown in Figure 2. It should be highlighted that full-text reading of the selected articles ($n$ = 15) was conducted. After that, the articles were divided into three major categories, namely, growth and fabrication, electrical properties, and structural, morphological, and optical properties. These categories were mainly linked to non-polar GaN-based photodetectors. The growth and fabrication were divided into several sub-categories, such as substrate use, photodetector geometry, contact layer geometry, layer geometry, and techniques. The electrical properties were sub-categorized into responsivity, detectivity (D) rise and fall time, noise equivalent power (NEP), applied bias, and light of illumination power density. Meanwhile, the last category was classified into high-resolution X-ray diffraction, Raman spectroscopy, transmission electron microscopy (TEM), scanning electron microscopy (SEM), atomic force microscopy (AFM), and photoluminescence (PL). These categories were based on the articles obtained from the six digital databases.

### 3.1. Growth and Fabrication

It is noteworthy that the architectures of photodetectors rely on the desired designs for various applications. This includes applications from basic electronics such as detectors and optical data storage devices towards more sophisticated applications such as semiconductor photodetectors for industry and safety and environmental monitoring [9,10,30,53]. This section discusses several important factors, such as epitaxially grown structures and the fabrication processes that played important roles in the novel design of non-polar GaN-based photodetector devices [3,9,30]. As mentioned earlier, several sub-criteria are discussed under the growth and fabrication of non-polar GaN-based photodetectors, namely, substrate use, photodetector geometry, contact layer geometry, and techniques based on the selected articles ($n$ = 15).

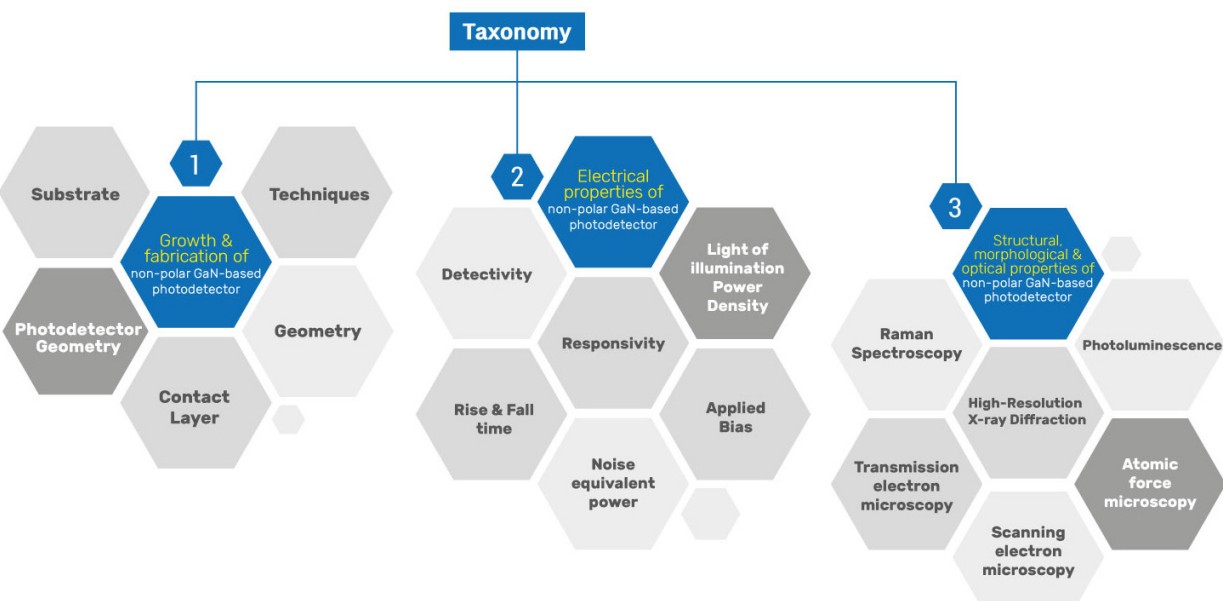

**Figure 2.** Taxonomy of the literature.

### 3.1.1. Substrate

Different substrates utilized for GaN-based photodetectors based on the selected articles according to the obtained articles, namely, c-plane sapphire, r-plane sapphire, a-plane sapphire substrate, and Si substrates, as shown in Figure 3.

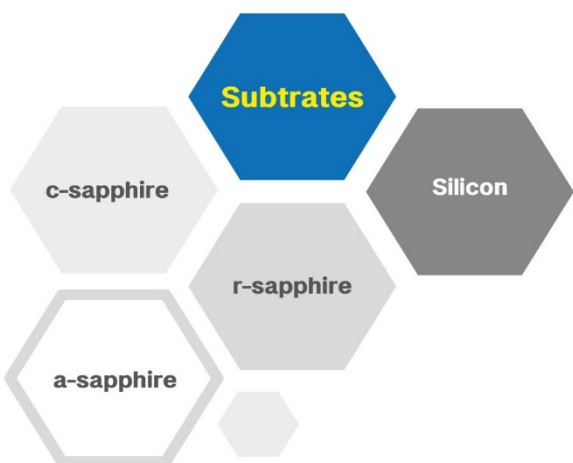

**Figure 3.** Block diagram of the substrates utilized for non-polar GaN-based photodetectors.

### I.    C-plane Sapphire Substrate

A growth of a GaN-based UV photodetector using the polar substrate along the [0001] direction was implemented by designing a 2-D heterojunction [54]. The fundamental of the piezo-phototronic effect as applied to a single non-polar a-axial GaN nanowire-based metal–semiconductor–metal (MSM) UV photodetector was obtained. In addition, the development of NWs using a c-plane sapphire substrate was reported by Zhang et al. [55]. Despite having growth in the vertical orientation (parallel towards the [0001] direction), the NWs were developed using an alloy such as InGaN/GaN core-shell, which showed radial growth. In turn, the growth was subjected to obtain non-polar m- or a-oriented directions; the InGaN/GaN core-shell could be deposited on the non-polar m-plane lateral facets of the wires owing to the three-dimensional structure. Therefore, the internal field in core/shell InGaN/GaN QWs was eliminated and the absorption efficiency was improved.

II. R-plane Sapphire Substrate

It is noteworthy that the common substrates to obtain GaN based-photodetectors are r-plane sapphire substrates [23]. Pant et al. reported the successful growth of GaN along the orientation of an r-plane sapphire substrate. Vast studies on a-plane GaN have also reported enhanced external quantum efficiency (EQE) and response and recovery times compared to polar c-plane GaN. In addition, due to the intrinsic polarization changes of Schottky barrier height that could hinder the photodetection capabilities of polar c-plane, a-plane GaN for UV photodetectors was preferable [19]. Moreover, three different growth approaches were employed for the growth of non-polar a-plane GaN-based photodetectors on r-plane sapphires in contrast to the conventional growth methods [30]. The EQE of the photodetectors fabricated in the (0002) polar and ($11\bar{2}0$) non-polar growth directions were compared accordingly.

Gundimeda et al. fabricated a non-polar GaN-based MSM UV photodetector [7]. The adaptation of an r-plane sapphire substrate for a-plane GaN growth uncovered new studies and techniques. More interestingly, the behavior of the a-plane GaN epitaxial layer towards the extended defects was investigated [31]. In particular, the dislocation densities and basal stacking faults (BSFs) that hindered the electrical performance of the a-plane GaN-based photodetector were greatly reduced. This approach was introduced using a commercially cheap r-plane sapphire substrate to promote laterally grown GaN devices along the non-polar orientation ($11\bar{2}0$). Another simple yet remarkable example using a non-conventional direction along the non-polar direction was the effect of using a high-related indium content [53]. The structures of non-polar ($11\bar{2}0$) a-plane InGaN epilayers grown on 200 nm $GaN/Al_2O_3$ ($1\bar{1}02$) substrate were obtained. An investigation of the photo-detecting properties was carried out. They reported that their devices were sensitive to both infrared and ultraviolet radiation. Despite the development of a fast response of the photodetector device towards the epitaxial growth, a study of in-plane anisotropic photoconduction of the non-polar a-plane GaN epitaxial layer was conducted [17]. The work comprised non-polar a-plane GaN epitaxial films grown on an r-plane sapphire. Another interesting work on photodetector devices is surface-engineered nanostructured non-polar ($11\bar{2}0$) GaN-based high-performance UV photodetectors [20]. In this work, four types of surface-engineered nanostructured along the non-polar direction were introduced and epitaxial a-plane GaN films were grown on an r-plane sapphire substrate.

III. A-plane Sapphire Substrate

Alternatively, non-polar a-plane sapphire substrate was utilized to grow c-plane GaN. The main interest of the aforementioned substrate was due to the minimal lattice mismatch between c-plane GaN grown on a-plane sapphire substrate, leading to the production of nano-porous (NP)- and nano-column (NC)-GaN features on sapphire ($11\bar{2}0$) substrate [56].

IV. Si Substrate

Despite the popularity of using sapphire substrate, motivation towards heteroepitaxially grown non-polar a-plane GaN was also extended to use an Si substrate. The work of Cai et al. reported the growth of a non-polar GaN stripe array-based MSM photodetector on patterned (110) Si substrates using two-step processes that led to a major improvement in the crystal quality [9]. It should be noted that the key to obtaining a high-crystalline-quality a-plane GaN was achieved by employing selective area growth using a dry- and wet-etching process. In turn, it potentially reduced the extended defects, reducing the contamination from gallium (Ga) meltback etching.

Furthermore, the work using patterned Si (110) substrate was introduced and benefited its piezo-potential distributed along the transverse orientation of a-axis GaN when experiencing a compressive/tensile strain along the axial orientation [57]. Tsai et al. also worked on the same principle as mentioned above [58], whereby Si substrate with no preferential orientation was utilized. The reason was to introduce GaN with a nanowire (NW) structure due to its high surface-to-volume ratio, which was applied to optoelectronic applications. Another recent work in NWs was the establishment of single and ensemble

NW photoconductors/photodiodes for realizing in-depth studies between these two methods [32]. The conventional Si (111) substrate was used as a template for the growth of these NW structures.

### 3.1.2. Photodetector Geometry and Metal Contacts

The designs of the photodetectors were categorized into four types of geometry, namely, MSM semiconductor, p-n junction, p-i-n junction, and hybrid-type geometry, as shown in Figure 4 [1,52–54].

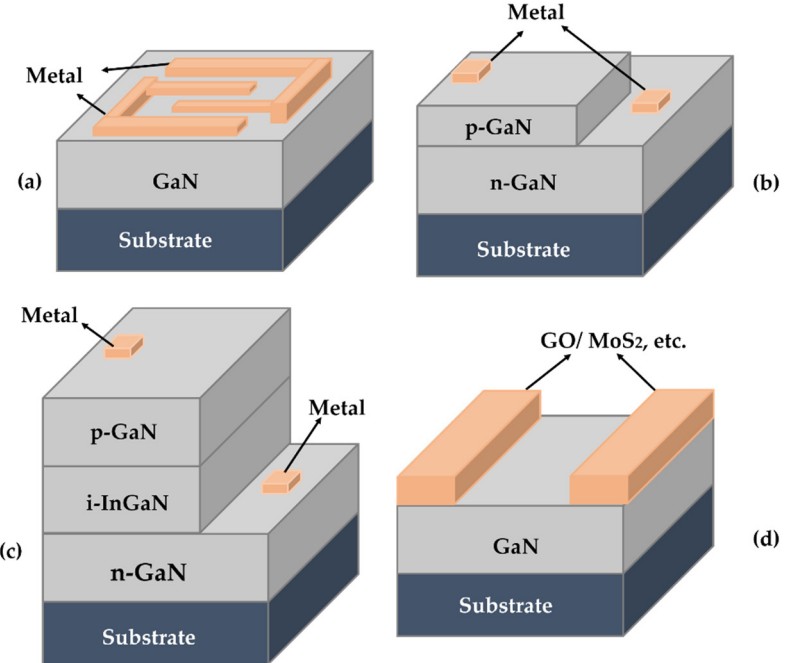

**Figure 4.** Schematic diagrams of non-polar GaN-based photodetectors with different designs: (**a**) MSM, (**b**) pn, (**c**) p-i-n, and (**d**) hybrid.

Due to its straightforward process, MSM geometry has been considered a popular geometry choice, especially in UV photodetectors that are fabricated on epitaxial GaN nanostructures [56]. MSM UV photodetectors were fabricated by depositing platinum (Pt) metal electrodes with a separation of 200 μm on GaN film. Gundimeda et al. fabricated a non-polar a-plane GaN-based MSM device structured for UV photodetectors [7]. The fabrication of this MSM UV photodetector was implemented using a gold (Au) contact with enhanced electrical performance. Meanwhile, the work of Cai et al. on non-polar GaN stripe arrays also featured the same type of GaN MSM photodetector [9]. The MSM photodetector was subsequently fabricated on non-polar a-plane GaN with a Schottky metal contact comprising Ti/Ai/Ti/Au (50/200/50/50 nm). It was discerned that the MSM structure comprised two symmetrical electrodes, namely, Ohmic and Schottky. Pant et al. fabricated MSM photodetectors using Au as electrodes, known as interdigitated electrode (IDEs), which provided a large excitation area compared to other available fabrication photodetector modules. This in turn resulted in an enhanced responsivity and fast transit time [3]. Tsai et al. also worked on the same principle as mentioned in the mechanism of the piezo-phototronic effect [19]. The fundamental of the piezo-phototronic effect as applied to a single non-polar a-axial GaN nanowire-based MSM UV photodetector was discussed. The NWs were fabricated into a single NW flexible MSM-type piezotronic device using silver paste at both ends of the GaN NW as a source and drain electrodes with Schottky junctions.

In addition, the electrical performance of a fabricated MSM a-plane GaN photodetector with Au/Ni as the metal contact was studied [31,32]. The mechanisms of the screw or mixed dislocation, edge dislocation, and BSF densities that affected the dark current, responsivity, and response time of GaN $(11\bar{2}0)$-based photodetectors were discussed. Another simple yet

remarkable approach using a non-conventional direction along the non-polar a-direction was the effect of consuming a highly related indium content [53]. The fabrication of this MSM photodetector device involved a metal of Al deposition and lift off to create the interdigitated electrode structures. In addition, the attempt to grow self-assembled non-polar high indium-InGaN quantum dots for the fabrication of a highly responsive MSM UV-photodetector was introduced [23]. The Au as the Schottky contact with GaN films was utilized. Despite the development of a fast response of photodetector devices, a study on the in-plane anisotropic photoconduction along the non-polar a-plane was conducted [23]. The geometry of the photodetector was MSM with the use of Au as the Schottky contact. Furthermore, the external quantum efficiency of the photodetectors fabricated along the (0002) polar and (11$\bar{2}$0) non-polar growth directions were compared accordingly [30]. In this work, the common design of the MSM structure was developed due to its simplicity, whereby the structure comprised two Schottky electrodes deposited on an unintentionally doped semiconductor layer. The fabrication of the device involved Au as metal contacts. On the other hand, the surface-engineered nanostructured non-polar (11$\bar{2}$0) GaN-based high-performance UV photodetectors were studied [20]. The photodetectors were fabricated using the MSM approach using Au metal contacts with decent electrical performance.

In contrast, the use of p-n junction geometry in NW design, which was perpendicular to the wire axis, was introduced [32]. However, the structure was not favorable due to the presence of additional leakage via surface states. It was reported that the p-n junction was designed by combining a p-type GaN NW/n-Si heterostructure. The devices were fabricated by depositing Au (100 nm)/Ni (25 nm) interdigitated electrodes (IDE) (active area of 12 mm$^2$) onto the NWs using thermal evaporation and establishing an electrical contact with the Si substrate. Basically, the study reported a comparative analysis of ensemble and single non-polar Mg-doped p-GaN NW and n-Si heterojunctions. In contrast, Zhang et al. proposed a method to prevent the additional leakage and a photoconductive element in the circuit that acted to perturb the photodiode operation [55]. Core-shell p-i-n InGaN/GaN NW photodetectors protected the junction from the surface and eliminated the photoconductive effect. The metal contact of Ti/Al/Ti/Au (10/20/10/200 nm) was deposited by using e-beam lithography for fabrication of the p-i-n NW photodiodes.

Another interesting work that should be taken into consideration embedded an organic polymer with an inorganic semiconductor nanomaterial such as GaN, ZnO, CdS, or Si. A promising hybrid heterostructure enabled the fabrication of a photosensor by implementing a photo-response of an a-axis GaN Microware (MW)/p-Polymer hybrid using the piezo-phototronic effect [57]. This hybrid heterostructure was deemed promising due to its easy fabrication and high flexibility of organic components with the superior electrical and optical properties of the inorganic semiconductor materials. To study the electrical properties of the hybrid device, the end of the MWs was fixed with silver paste. In contrast, a design in 2-D heterojunction-based UV photodetectors enabled a 2D/3D-based heterostructure and its applications in highly sensitive and quick UV photodetectors with excellent performance [54]. The MoS2/c-GaN-based heterojunction was actualized through a two-step process, namely, the deposition of a highly controllable and large scalable Mo film by DC-sputtering on top of the GaN substrate, followed by sulfurization. The fabrication involved a deposition of Al/Au for the metal contact of the device.

### 3.1.3. Deposition Technique and Layer Geometry

Other significant sub-criteria to highlight are the layer geometry and deposition techniques, as listed in Table 1. The deposition methods of the photodetectors relied on various available techniques such as plasma-assisted molecular beam epitaxy (PAMBE), metal organic chemical vapor deposition (MOCVD), laser molecule beam epitaxy (LMBE), chemical vapor deposition (CVD), and atmospheric pressure chemical vapor deposition (APCVD), as illustrated in Figure 5. These techniques are promising and could match the current industry practice in producing high-performance devices as well as enhancement of the crystal quality of the heteroepitaxial layer.

**Table 1.** Summary of the growth and fabrication of non-polar GaN-based photodetectors.

| Substrate | | | | Photodetector Geometry | | | | Contacts | | | | | | | Layer Geometry | | | Technique | | | | | | | Ref |
|---|---|---|---|---|---|---|---|---|---|---|---|---|---|---|---|---|---|---|---|---|---|---|---|---|---|
| c-sap | r-sap | Si | a-sap | MSM | pn | p-i-n | Hybrid | Al/Au | Au/Ni | Au | Ti/Al/Ti/Au | Pt | Silver Paste | Al | TF | NS | MS | PAMBE | PECVD | MOCVD | LMBE | CVD | APCVD | PLD | |
| - | √ | - | - | √ | - | - | - | - | - | √ | - | - | - | - | √ | - | - | √ | - | - | - | - | - | - | [7] |
| - | - | √ | - | √ | - | - | - | - | - | - | - | - | - | - | √ | - | - | - | √ | √ | - | - | - | - | [9] |
| - | √ | - | - | - | - | - | - | - | - | √ | - | - | - | - | √ | - | - | √ | - | - | - | - | - | - | [19] |
| - | - | - | √ | √ | - | - | - | - | - | - | - | √ | - | - | √ | - | - | - | - | - | √ | - | - | - | [56] |
| - | - | √ | - | - | - | - | √ | - | - | - | - | - | √ | - | - | - | √ | - | √ | √ | - | - | - | - | [57] |
| - | - | √ | - | √ | - | - | - | - | - | - | - | - | √ | - | - | √ | - | - | - | - | - | - | √ | - | [58] |
| √ | - | √ | - | - | - | √ | - | - | - | √ | - | - | - | - | - | √ | - | - | - | √ | - | - | - | - | [55] |
| - | - | √ | - | - | √ | - | - | - | √ | - | - | - | - | - | - | √ | - | - | - | - | - | - | √ | - | [32] |
| √ | - | - | - | - | - | - | √ | √ | - | - | - | - | - | - | √ | √ | - | √ | - | - | - | - | - | - | [54] |
| - | √ | - | - | √ | - | - | - | - | √ | - | - | - | - | - | √ | - | - | - | - | √ | - | - | - | √ | [31] |
| - | √ | - | - | √ | - | - | - | - | - | √ | - | - | - | - | √ | - | - | - | - | - | √ | - | - | - | [30] |
| - | √ | - | - | √ | - | - | - | - | - | - | - | - | - | √ | √ | - | - | √ | - | - | - | - | - | - | [53] |
| - | √ | - | - | √ | - | - | - | - | - | √ | - | - | - | - | √ | - | - | - | - | - | - | √ | - | - | [17] |
| - | √ | - | - | √ | - | - | - | - | - | √ | - | - | - | - | √ | - | - | √ | - | - | - | - | - | - | [23] |
| - | √ | - | - | √ | - | - | - | - | - | √ | - | - | - | - | √ | √ | - | √ | - | - | - | - | - | - | [20] |

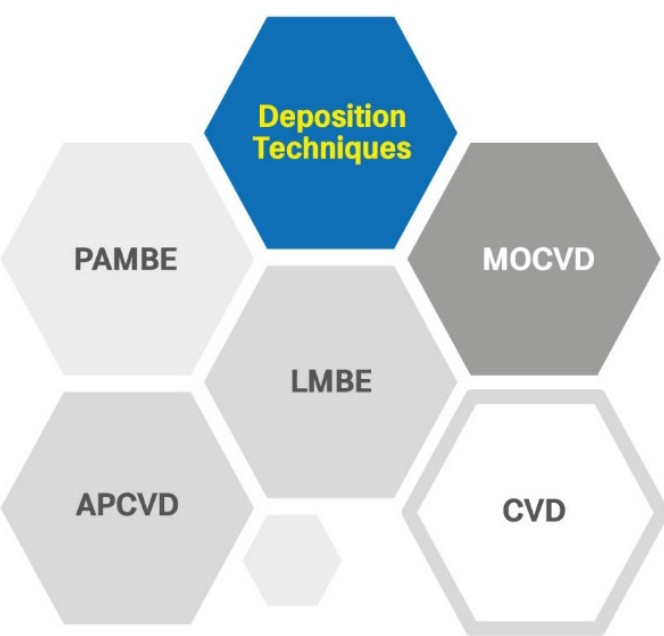

**Figure 5.** Block diagram of the techniques utilized for the growth of non-polar GaN-based photodetectors.

I.    Plasma-assisted Molecular Beam Epitaxy

The utilization of plasma would assist the growth by supplying rich active nitrogen species in the chamber [19]. PAMBE is a well-known crystal growth deposition technique that uses an ultra-high vacuum condition, leading to high crystallinity of the epitaxial layers. The growth of a high-quality non-polar a-plane GaN epitaxial layer with a minimal set of pre-growth conditions using PAMBE was introduced [19]. Three different growth approaches were employed for the growth of non-polar a-plane GaN in contrast to the conventional growth techniques. Gundimeda et al. reported the growth of triangular island-like structures on the surface of a-plane GaN with a high crystalline and stress-free epitaxial layer using PAMBE [7].

Furthermore, a highly controllable deposition and large scalable Mo film by DC-sputtering on top of GaN thin film followed by sulfurization was reported using PAMBE [55]. The design in the 2-D heterojunction-based UV photodetector enabled the 2D/3D-based heterostructure [54]. In turn, this approach led to applications in highly sensitive and quick UV photodetection. Besides, self-assembled non-polar high indium clusters of $In_{0.55}Ga_{0.45}N$ grown on a non-polar $(11\bar{2}0)$ a-plane In0.17Ga0.83N epilayer using the PAMBE system were introduced [53].

On the other hand, a report of a-plane GaN growth on r-sapphire with IDE patterns of Au were fabricated to restrict the carrier transport along the [0002] direction using the PAMBE system; this approach reported a highly responsive UV photodetector [23]. A study of the in-plane anisotropic photoconduction of non-polar epitaxial a-plane GaN was conducted using the PAMBE system [17]. Another interesting work on photodetector devices is a surface-engineered nanostructure non-polar $(11\bar{2}0)$ GaN-based high-performance UV-photodetector [20]. The epitaxial a-plane GaN thin films were grown on an r-plane sapphire substrate by the PAMBE system. The GaN film exhibited a nanostructure with a pyramidal structure (PS), flat nanorods (FN), a trigonal shape structure (TSS), and trigonal prism-shaped nanorods (TPN) in GaN-based UV-photodetectors.

II.    Metal Organic Chemical Vapor Deposition

Another possible technique to obtain non-polar GaN-based photodetectors is MOCVD. Cai et al. reported the growth of non-polar GaN stripe array-based photodetectors on patterned (110) Si substrates [9]. The growth was obtained by using two-step-processes, which led to a major improvement in the crystal quality. The preparation of patterned Si

(110) substrate was implemented by depositing $SiO_2$ using the plasma-enhanced chemical vapor deposition (PECVD) technique. After the desired patterned was obtained, the a-plane GaN was epitaxially grown using MOCVD. Song et al. reported that the preparation of epitaxial growth in hybrid photodetectors involved a two-step approach, namely, the PECVD and MOCVD systems [57]. They reported the preparation of patterned Si substrate by PECVD, followed by the epitaxial growth of GaN microwires with the use of an Aixtron $3 \times 2''$ showerhead MOCVD reactor (Thomas Swan Scientific Equipment Ltd.). Apart from that, the effect of the dislocations and BSFs of the epitaxial layer grown by MOCVD towards the performance of a-plane GaN-based photodetectors in terms of electrical properties was reported [31]. Another significant finding of catalyst-free n-doped GaN NWs grown by MOCVD was reported by Zhang et al. [55]. They reported the use of non-polar InGaN/GaN QWs as the cover for the top part of the wire with a p-doped GaN outer shell. It is noteworthy that with the ability to produce a mass scale-up production (depending on the capabilities of the industries, i.e., 4- and 8-inch wafers), MOCVD lives up to its reputation as being one of the best deposition methods, especially in yielding a high-crystal-quality epi-structure.

III. Laser Molecule Beam Epitaxy

The use of lasers has been reported with molecular beam epitaxy (MBE) for the epitaxial growth of GaN film and NP- and NC-GaN on sapphire (11$\bar{2}$0) substrate [56]. They reported that the desired layer geometry could be achieved by tuning the buffer layer condition. The GaN film was obtained in 2D epitaxial layers with the subsequent growth of NC and NP alternating on top of the 2D GaN layer. Furthermore, non-polar (11$\bar{2}$0) GaN epilayers were grown on (1-102) r-plane sapphire substrates by varying the nitrogen flow using MBE. In this work, the optimized quality non-polar (11$\bar{2}$0) a-plane GaN grown on (1$\bar{1}$02) r-plane sapphire substrate was implemented by optimizing the growth conditions [30].

IV. Chemical Vapor Deposition

NWs were also reported by Tsai et al. using the CVD system, whereby Au was used as a catalyst during NW growth [58]. In contrast, despite the essential utilization of catalyst to produced NW structures, Zhang et al. showed NWs of InGaN/GaN core shell without the presence of a catalyst inside the MOCVD reactor [55].

V. Atmospheric Pressure Chemical Vapor Deposition

Recent work on single and ensemble NW photoconductors and photodiodes to realize in-depth studies between these two methods was reported [32]. The growth of the NW structure was reported using the APCVD technique via the catalytic vapor–liquid–solid (VLS) process.

As for the growth and fabrication of non-polar GaN-based optoelectronics, several categories have been reported, namely, substrate, photodetector geometry, contacts, layer geometry, and technique. As can be seen from Table 1, the use of Si substrates was reported by five research groups, whereas c-plane sapphire substrates and a-plane sapphire substrates were reported by two research groups and one, respectively. Meanwhile, the use of r-plane sapphire substrates was reported to be utilized by eight research groups. In turn, this shows the importance of r-plane sapphire substrates and their impact on achieving the growth of a non-polar GaN epitaxial layer.

In contrast, the use of MSM structurers as photodetector geometry has been reported by 10 research groups compared to pn, p-i-n, and hybrid structures. It is noted that the type of MSM structure has been extensively utilized due to several advantages such as simpler fabrication process and lower capacitance compared to other structures. In addition, the MSM structure comprises Schottky contacts, which can lead to a low dark current, high internal gain, and reduced noise [23,59–61].

On the other hand, the use of thin film as a layer geometry has been widely utilized compared to other structures such as NS and MS due to the simplicity of the growth proce-

dures (in-situ techniques). However, it is noteworthy that the growth of NS and MS could lead to great enhancement due to the reduction of defects and increment of photosensitivity.

Another category to highlight is the techniques that were utilized to grow the epitaxial layers, such as PAMBE, MOCVD, LMBE, CVD, and APCVD. It can be observed that the use of PAMBE obtained the priority to grow non-polar GaN-based photodetectors, whereas CVD and APCVD were reported to be utilized the least. It is noteworthy that the growth of GaN epitaxial layers was achieved with high crystal quality due to the ultra-high vacuum (UHV) condition obtained in MBE compared to other techniques [62]. In turn, it was discerned that the MBE systems could promote a control on the film thickness down to fractions of monolayers.

### 3.2. Electrical Properties

Here, the sub-criteria of the electrical properties of non-polar GaN-based photodetectors are discussed based on the selected articles ($n$ = 15).

### 3.2.1. Responsivity

It is noteworthy that the photodetectors should fulfil various requirements to ensure that their working principle align with the device efficiency in detecting light. The electrical properties play a prominent role in understanding how the device works as a photon detector, which realizes the photoexcitation of electric carriers. It should also be noted that the ability to sense a specific range of wavelength, responsivity, and radiant sensitivity are key for photodetector devices. It is seen that non-polar nanostructure GaN epitaxial layers portrayed lower qualities compared to conventional polar nanostructures [30]. In addition, the crystallographic orientation and surface polarity were seen to play a crucial role in determining the performance of the photodetectors by modifying the metal–semiconductor contact. A non-polar a-plane GaN-based photodetector with a wavelength of 360 nm was reported by Mukundan et al. [30]. The photoresponsivity exhibited 0.155 A/W for the non-polar a-plane GaN and 0.00567 A/W for the polar c-GaN. Another simple example using a non-conventional direction along the non-polar direction was the effect of consuming a highly related indium content [53]. The structure was grown and investigated for photodetection properties, which showed a sensitivity to both infrared and ultraviolet radiation. For an incident wavelength of 360 nm at 2V applied bias, the measured responsivity was 0.082 A/W for the UV illumination, and for an incident wavelength of 1000 nm (broad source) at 2 V, the responsivity was $9.57 \times 10^{-4}$ A/W for the infrared (IR) illumination. In addition, a study reported by Yang et al. found that the screw or mixed dislocation increased the dark current mainly by reducing the Schottky barrier height and forming a leakage current. This was mainly due to the great reduction in the responsivity by reducing the electron mobility that resulted from the edge dislocations and BSFs [31]. The responsivities were 0.61, 0.82, 1.04, 1.31, and 1.60 A/W, ranging from 1 to 5 V.

In addition to the aforementioned, a spectral response was conducted by varying the bias voltage from 0 to 5 V on the work of highly responsive and self-powered a-plane GaN-based UVA photodetectors [17]. In this work, the highest responsivity was obtained from the first sample of 400 mA/Wat 5 V, whereas at 0, 1, 2, 3, and 4 significant values of 4.67 mA/W, 25.3 mA/W, 42.1 mA/W, 81.4 mA/W, and 165 mA/W were shown, respectively. A study of a-plane GaN also reported that the EQE and the response and recovery times of non-polar GaN were more efficient compared to polar c-plane GaN [23]. The response time, responsivity, and internal gain were 210 ms, 1.88 A/W, and 648.9%, respectively. Another interesting work on photodetector devices studied surface-engineered nanostructure non-polar (11$\bar{2}$0) GaN-based high-performance UV photodetectors [20]. The calculated responsivity values of PS, FN, TSS, and TPN GaN film-based UV photodetectors were 4.4, 5.6, 24.5, and 25.9 ($\pm$0.3) mA/W, respectively, with an applied bias of 3 V.

Another study showed that the fabricated GaN-based UV photodetector exhibited a high responsivity of 340 mA/W at 5 V bias at room temperature a with 25 times enhancement [7]. The increase in responsivity at different bias was also reported for non-polar (11$\bar{2}$0)

GaN-based MSM photodetectors [9]. The device performance exhibited a responsivity of 695.3 A/W at 1 V bias, 2218.53 A/W at 2 V bias, and 12628.3 A/W at 5 V bias, with both under 360 nm ultraviolet illumination. These finding were 20 times higher and four orders of magnitude higher than the current state-of-the-art photodetectors. Similarly, a growth of non-polar a-plane GaN for photodetection was also reported at a responsivity of 25 A/W at a bias of 1 V, under dark and illuminated conditions (UV-A, 364 nm, 0.06 mW/cm$^2$) [19]. The I-V characteristics were plotted for sweep bias from $-1$ V to 1 V for the samples, wherein all samples portrayed a proper response to UVA (364 nm) light. The exact values of the rise and fall time were not clearly reported, whereas the response time was in the sub-second range and the recovery time was <2 s. In addition, the responsivity towards the epitaxial growth of the GaN nanostructure was studied for photodetector applications [56]. Three devices were fabricated, and it was revealed that the non-polar GaN-based MSM photodetector had a photo-response of ~358 mA/W at an applied bias of 1 V. It was suggested that the photo-responsivity of the non-polar GaN-based MSM photodetector was higher than that of GaN film (~36 mA/W) and NC-GaN (~7 mA/W). In turn, it revealed the importance of the shape and the size of GaN nanostructures on the responsivity of UV photodetector devices.

Furthermore, the implementation of the piezo-phototronic effect was used by modulating the photo-response performances of a-axis GaN microwire/p-polymer inorganic/organic hybrid photosensors [57]. It resulted in a responsivity as high as $1.14 \times 10^4$ A/W and $7.5 \times 10^6$% under compressive strain. Interest in GaN NW has also increased due to its high surface-to-volume ratio of NWs, which have attracted great attention in photodetection applications. An ultrahigh UV responsivity of $1.3 \times 10^5$ A/W under an illumination intensity of 2.46 mW/cm$^2$ was reported for a single non-polar a-axial GaN-based UV photodetector incorporating the piezo-phototronic effect [58]. Besides that, a single p-n junction NW photodetector containing 30 non-polar radial InGaN/GaN QWs recorded a responsivity of 0.14 A/W at 3.35 eV for photo-responsivity [55]. Under reverse bias of $-1$ V, the spectral shape remained similar, whereas the photocurrent signal increased, and the peak responsivity was 0.157 A/W. On the other hand, a comprehensive study between ensemble and single NW devices on non-polar p-GaN/n-Si heterojunction diode characteristics showed an enhancement of five orders at a wavelength of 470 nm [32]. The ideality factor of the single NW p-GaN/n-Si device was found to be three times lower than the ensemble NW device. The responsivity of photodiode characteristics of the ensemble NW heterojunction device portrayed that the maximum responsivity was 69 mA/Wat 470 nm, whereas the responsivity was 18 mA/W at 530 nm. In contrast, a photodiode characteristic of a single NW heterojunction device recorded the maximum responsivity of 1675 A/W at 470 nm and 197 A/W at 530 nm. These high responsivity values yielded high photocurrent gains. A great breakthrough was also found in the work on wafer-scale synthesis of a uniform film of MoS2 with few layers on GaN for a 2D heterojunction UV photodetector [54]. The device performance portrayed a high external spectral responsivity of up to ~$3.05 \times 10^3$ A/W with ultrahigh responsivity.

3.2.2. Detectivity Jones and Noise Equivalent Power (NEP)

Despite the importance of the responsivity, the performance of photodetectors was also determined by attaining proper detectivity (D). The NEP and D are also two key figures of merit parameters to determine the performance of photodetectors. They could correspond to the ability of the detector to detect the weakest light signals. The D and NEP are expressed by the following equation [7]:

$$D = \frac{(A\Delta f)^{\frac{1}{2}}}{NEP} \tag{1}$$

$$NEP = \frac{\overline{I_n^2}^{\frac{1}{2}}}{R} \tag{2}$$

where A, Δf, and $\overline{\mathrm{l}^2_n}^{\frac{1}{2}}$ are the device area, the electrical bandwidth, and the measured noise, respectively.

A device reported by Gundimeda et al. exhibited a small value of NEP of $2.4 \times 10^{-11}$, which was less than the existing silicon diode [7]. Such a low NEP value was possible due to the small noise currents at the low bias. The D was then calculated to be $1.24 \times 10^9$ Jones at 5 V. Based on the equation above, Aggarwal et al. reported the calculated D of the MSM photodetector of GaN film and NP- and NC-GaN values of $1.57 \times 10^8$ Jones, $1.01 \times 10^8$ Jones, and $2.46 \times 10^8$ Jones, respectively [56]. In addition, the NEP values for GaN film and NP- and NC-GaN were $6.59 \times 10^{-10}$, $7.16 \times 10^{-10}$, and $2.76 \times 10^{-10}$ W·Hz$^{-1/2}$, respectively. Apart from that, a 2D/3D heterojunction-type photodetector utilizing an MoS2 heterojunction on GaN substrate with a mass-scalable sputtering method portrayed a D of ~$10^{11}$ Jones under UV illumination of varying intensities [55]. The highest D of a photodetector device was reported in a self-powered a-plane GaN-based UVA photodetector driven by unintentional asymmetrical electrodes [17]. The D was claimed to be $3.0 \times 10^{13}$ Jones, slightly higher than GaN thin film-based devices. This value was the highest compared to other works reviewed in this literature. In addition, a surface-engineered nanostructure non-polar ($11\overline{2}0$) GaN-based high-performance UV photodetector demonstrated significant results in the D and the NEP [20]. Each nanostructured type exhibited a variation in the detectivity values. The calculated D values of PS, FN, TSS, and TPN GaN film-based UV photodetectors were $0.43 \times 10^8$, $0.76 \times 10^8$, $2.73 \times 10^8$, and $2.83 \times 10^8$ Jones, respectively. The low NEP in the order of $10^{-10}$ W·Hz$^{-1/2}$ indicated a very low noise power in the devices, which promoted suitable candidates for highly sensitive GaN-based UV photodetectors. The calculated NEP for PS, FN, TSS, and TPN nanostructures were $31.7 \times 10^{-10}$, $17.5 \times 10^{-10}$, $4.8 \times 10^{-10}$, and $4.7 \times 10^{-10}$ W·Hz$^{-1/2}$, respectively.

### 3.2.3. Rise and Fall Time

Another important characteristic of the photodetection application is the response time. The response time is the time it takes for the detector output to change in response to changes in the input light intensity. This time can be deliberated into the two most common terms, namely, the rise and the fall times. A non-polar a-plane GaN-based UV photodetector with an enhanced response time, claimed to be the fastest response, was reported by Gundimeda et al. [7]. At a bias voltage of 5 V, the calculated rise time and fall time were 280 ms and 450 ms, respectively. This suggests that the value of these times in milliseconds was enhanced compared to the previous reports in seconds. The enhanced photo-response was mainly attributed to low trap density and the high crystalline quality of the GaN film. Another research group reported a non-polar photodetector device with improved results in the rise and fall time [9]. By using a standard exponential fitting, the rise time and the fall time of the MSM photodetector were 66 and 43 μs, respectively, which were three orders of magnitude faster than those using the current state-of-the-art non-polar GaN photodetectors. In addition, the response time of a photodetector based on an optimized sample condition was depicted accordingly [31]. Under higher bias, the separation of the photon-generated carrier could be promoted, increasing the photocurrent, which could lead to enhanced responsivity. The photo and dark currents were 8.5 and 1.1 nA, respectively, whereas the rise and fall times were 24 and 8 ms, respectively.

In contrast, the work on self-powered and highly responsive non-polar a-plane GaN-based UV photodetectors portrayed great findings [17]. However, the values of transit times for gate and drain were at 0.05 and 0.12 s, respectively. These values were slightly higher than the other types in non-polar GaN UV photodetectors. In addition to the aforementioned, the study on UV detection performance along the anisotropic directions in non-polar based MSM photodetectors was investigated [23]. It should be highlighted that the response time was improved along the azimuths as [0002] > [1-102] > [1-100], which portrayed the unanimous effect of azimuth angles on non-polar a-plane GaN film, although the recovery time remained constant. The values for rise and fall times were 0.21 and 1.2 s, respectively. Surface-engineered nanostructure-based non-polar GaN MSM high-

performance photodetectors were demonstrated for the detection of UVA radiation [20]. It is noteworthy that the lowest rise and fall times for TPN-based UV photodetectors led to a fast response of the fabricated UV photodetectors after exposure to UV radiation along with a quick recovery time. It was also observed that the fall time was decreased in FN (1118 ms). However, it further increased for TSS (1520 ms), and ultimately the lowest value was achieved for TPN (995 ms). Fitted values of the rise and fall time constants ($\tau r$ and $\tau d$) were 853 ms and 1622 ms, 45 ms and 509 ms, 180 ms and 692 ms, and 151 ms and 453 ms for PS, FN, TSS, and TPN nanostructures, respectively.

3.2.4. Light of Illumination Power Density and Applied Bias

To further determine the photocurrents generated in photodetectors, the illumination of a light source such as a focus laser and UV light is utilized. Typically, the spectrometer is equipped with a desired wavelength with power densities varying from 0 to 100 W/cm$^2$ depending on the suitable power density of the measurement. For instance, in performing the non-polar GaN-based highly responsive and fast UV photodetector, the spectrometer used in that setup featured a focused laser of $\lambda$ = 325 nm and power density = 1–13 mW/cm$^2$ [7]. The UV light power density used in this work was 13 mW/cm$^2$. Another research group reported a higher light power density of 500 mW/cm$^2$ [9]. The non-polar ($11\bar{2}0$) GaN-based MSM photodetectors utilized a power density between 0.15–0.55 W/cm$^2$. Moreover, another research group reported the use of an output power of 0.06 mW/cm$^2$ via the use of a 300 W xenon bulb that provided a range of wavelengths from 200 to 1200 nm [19]. This work reported an improved quality of non-polar a-plane GaN thin films by introducing unconventional new efficient growth conditions without compromising their UV photodetection properties. Additionally, high responsivity was achieved via different applied bias voltages (0.2–3 V) and optical laser powers (1–13 mW/cm$^2$) [56]. Breakthrough results were reported using the hybrid-type photosensor [58]. The illuminating intensities of 0.14 mW/cm$^2$ and 1.1 mW/cm$^2$ were used along with an enhanced sensitivity and responsivity of the device of 508% and 354%, respectively.

In contrast, piezotronic and piezo-phototronic measurements were performed for ultrahigh UV responsivity of a single non-polar a-axial GaN NW [58]. The piezo-phototronic effect was studied via the influence of photon excitation. A He-Cd laser with a central peak wavelength at 325 nm was applied as an excitation source. A neutral density (ND) filter was used to change the excitation intensity for different measuring conditions of seven excitation intensities (0, 2.46, 4.92, 9.84, 19.68, 29.52, and 39.36 mW/cm$^2$). Another type of hybrid 2-D heterojunction UV photodetector was fabricated and the behavior of the device under 365 nm light irradiation was studied [57]. The light modulation density in this work used up to 100% and the bias was between 0 V and 1 V. Another study reported the UV illumination of various intensities from 10 to 70 mW/cm$^2$ for a 2D heterojunction UV photodetector, and the voltage bias was fixed at 1 V [54]. Meanwhile, an intensity of 0.3 mW/cm$^2$ with a wavelength of 360 nm was utilized in the photo-response study with an applied bias fixed at 2 V [30]. Apart from that, the EQE of the photodetectors fabricated along the (0002) polar and ($11\bar{2}0$) non-polar growth directions was introduced [53], whereby a lamp with a wavelength of 360 nm with an intensity of 0.3 mW/cm$^2$ was used as a UV light source. The same wavelength of 360 nm with an intensity of 0.3 mW/cm$^2$ was used as the UV light source by Mukundan et al. [17]. The results indicate that the epitaxial grown structure was sensitive to both infrared and ultraviolet radiations due to the different compositions of the InGaN region. In addition, a method of calculation was implemented to find the value of spectral responses for each wavelength as follows:

$$R = \frac{I_{light} - I_{dark}}{SL_{light}} \tag{3}$$

where $I_{light} - I_{dark} = \Delta I\lambda$ is the difference between the photo and dark current.

For highly responsive and self-powered a-plane GaN-based UVA photodetectors, the $I_{light}$ is the power of the UV source (0.06 mW/cm$^2$) and $SL_{light}$ is illuminated device

area (1 mm × 1 mm) [23]. Additionally, Pant et al. reported that the UVR and internal gain (G) were dependent on the azimuth angle [17]. The temporal response was very stable irrespective of growth conditions and azimuth angles. To determine the temporal response, a UV source of 360 nm wavelength was used along with a power density of 0.3 mW/cm$^2$ with an applied bias of 1.0 V and 5.0 V. In contrast, an enhanced photocurrent could be observed under UV illumination at wavelength of 325 nm with a power density of 13 mW/cm$^2$ [20]. This remarkable enhancement was an indicator of the setup parameters of light illumination and bias voltage used from 1.0 V to 4.0 V in the mentioned experiment.

It is noteworthy that several analyses have been reported to emphasize the performance of non-polar GaN-based photodetectors. The performance of the photodetector is based on several parameters, such as responsivity, D, rise and fall time, NEP, and light of illumination power density. It should be noted that the importance of the responsivity analysis has gained huge attention in deciding on the device performance. As can be unambiguously discerned from Table 2, the highest responsivity of $1.3 \times 10^5$ (A/W) was obtained by Tsai et al. [58], whereas the lowest responsivity of 1.8803 mA/W was reported by Pant et al. [23]. In addition, another important analysis to mentioned is D. Based on the targeted articles, several research groups reported the values of D, with the highest value of D reported by Pant et al. [17], of $3.0 \times 10^{13}$ Jones, and the lowest values of D reported by Mishra et al. ($0.43 \times 10^8$ Jones) [20]. In contrast, the rise and fall times were reported to be utilized to demonstrate the photodetector device efficiency. It could be seen that the fastest response of rise and fall times reported by Cai et al. were 66 and 43 μs, respectively [9]. For the NEP of non-polar GaN-based photodetectors, the lowest value was obtained with $2.4 \times 10^{-11}$ W·Hz$^{-1/2}$, as reported by Gundimeda et al. [7].

**Table 2.** Summary of electrical properties.

| Responsivity | Detectivity Jones | Rise and Fall Time | Noise Equivalent Power $W \cdot Hz^{-1/2}$ | Applied Bias | Light of Illumination Power Density | Ref. |
|---|---|---|---|---|---|---|
| 340 mA/W | $1.24 \times 10^9$ | 280 ms and 450 ms | $2.4 \times 10^{-11}$ | 5 V | 1–13 mW/cm$^2$ | [7] |
| 695.3 A/W<br>2218.53 A/W<br>12,628.3 A/W | - | 66 and 43 µs | - | 1 V<br>2 V<br>5 V | ~0.15–0.55 W/cm$^2$ | [9] |
| 25 A/W | - | - | - | 1 V | 0.06 m W/cm$^2$ | [19] |
| NP-GaN (~358 mA/W)<br>GaN film (~36 mA/W)<br>NC-GaN (~7 mA/W) | $1.57 \times 10^8$<br>$1.01 \times 10^8$<br>$2.46 \times 10^8$ | - | $6.59 \times 10^{-10}$<br>$7.16 \times 10^{-10}$<br>$2.76 \times 10^{-10}$ | 1 V<br>1 V<br>1 V | 1–13 mW/cm$^2$ | [56] |
| $1.14 \times 10^4$ A/W | - | - | - | 3 V | 0.14–1.10 mW/cm$^2$ | [57] |
| $1.3 \times 10^5$ (A/W) | - | - | - | −2 V–+2 V | 2.46–39.36 mW/cm$^2$ | [58] |
| 0.14 A/W<br>0.16 A/W | - | - | - | 0 V<br>1 V | From 0 to 100% | [55] |
| ensemble NW 69 mA/W<br>single NW 1675 mA/W | - | - | - | ±4 V | - | [32] |
| $3.05 \times 10^3$ A/W | $10^{11}$ | - | - | 1 V | 10 to 70 mW/cm$^2$ | [54] |
| 0.61 A/W<br>0.82 A/W<br>1.04 A/W<br>1.31 A/W<br>1.60 A/W | - | 24 ms and 8 ms | - | 1 V<br>2 V<br>3 V<br>4 V<br>5 V | 0–10$^4$ W/m$^2$ | [31] |
| 0.155 A/W | - | 6 s and 15 s | - | 2 V | 0.3 mW/cm$^2$ | [30] |
| $9.57 \times 10^{-4}$ A/W | - | 3 s and 15 s | - | 2 V | 0.3 mW/cm$^2$ | [54] |
| 4.67 mA/W | $3.0 \times 10^{13}$ | 0.050 s and 0.12 s | - | 0 V | 0.06 mW/cm$^2$ | [17] |
| GaN375 = 1.8803 mA/W<br>13.024 mA/W | - | 0.21 s and 1.2 s | - | 1 V<br>5 V | 0.3 mW/cm$^2$ | [23] |
| Ps = 4.4 mA/W<br>FN = 5.6 mA/W<br>TSS = 24.5 mA/W<br>TPN = 25.9 mA/W | $0.43 \times 10^8$<br>$0.76 \times 10^8$<br>$2.73 \times 10^8$<br>$2.83 \times 10^8$ | 853 ms and 1622 ms<br>545 ms and 509 ms<br>180 ms and 692 ms<br>151 ms and 453 ms | $31.7 \times 10^{-10}$<br>$17.5 \times 10^{-10}$<br>$4.8 \times 10^{-10}$<br>$4.7 \times 10^{-10}$ | 3 V | 13 mW/cm$^2$ | [20] |

### 3.3. Structural, Morphological, and Optical Properties

The structural, morphological, and optical properties of non-polar GaN-based photodetectors are discussed based on the selected articles (n = 15), as listed in Table 3. In addition, each criterion is discussed based on sub-criteria to demonstrate the usability of each instrument towards the quality of the epitaxial layers.

**Table 3.** Summary of structural, morphological, and optical properties.

| Structural Properties | | | | | | Surface Morphology | | | Optical Properties | Refs |
|---|---|---|---|---|---|---|---|---|---|---|
| High Resolution X-ray Diffraction | | | | Raman Spectroscopy | TEM | SEM | AFM | | PL | |
| PA | On | Off | RSM | | | | | | | |
| √ | √ | - | - | √ | - | √ | - | | √ | [7] |
| - | √ | - | - | - | - | √ | - | | √ | [9] |
| √ | √ | - | - | √ | - | - | - | | - | [19] |
| √ | √ | - | - | √ | - | √ | - | | - | [56] |
| - | - | - | - | √ | √ | √ | - | | √ | [57] |
| - | - | - | - | - | √ | √ | - | | - | [58] |
| - | - | - | - | - | √ | √ | - | | √ | [55] |
| - | - | - | - | - | - | √ | - | | √ | [32] |
| √ | - | - | - | √ | - | √ | √ | | - | [54] |
| √ | √ | - | - | - | √ | - | √ | | - | [31] |
| √ | √ | - | √ | √ | √ | √ | √ | | - | [30] |
| √ | - | - | - | - | - | √ | - | | √ | [53] |
| √ | √ | - | - | - | - | √ | √ | | - | [17] |
| √ | √ | √ | √ | - | - | √ | - | | - | [23] |
| - | - | - | - | - | - | √ | - | | - | [20] |

#### 3.3.1. Structural Properties

The structural properties of the non-polar GaN are discussed based on the measurements of X-ray diffraction (XRD), Raman spectroscopy, and TEM. Different studies reported the use of the abovementioned measurements to evaluate the qualities of the epitaxial layers grown on different substrates.

I.   X-ray Diffractions

It is noted that the phase analysis measurement through XRD plays an important role in determining the nominal plane orientation of the grown GaN epitaxial layer, especially in non-polar epitaxial layers [62,63]. Studies have shown that most of the fabricated non-polar GaN-based photodetectors underwent the XRD phase analysis in the epitaxial GaN stage prior to the device fabrication process. The growth of non-polar a-plane GaN on r-plane sapphire substrate usually exhibited a high reflection of 2-theta scan at an angle of 25.5°, 52.5°, and 57.7°, which corresponded to r-plane (10-12) sapphire, $(20\bar{2}4)$ sapphire, and a-plane $(11\bar{2}0)$ GaN, respectively [7,17,19,23,30,31,53]. In contrast, devices using InGaN epitaxial layers with different indium compositions recorded peak positions at 54.27° and 56.65° for InGaN with higher In and Ga compositions, respectively [54]. Aggarwal et al. reported the growth of c-plane GaN on a-plane $(11\bar{2}0)$ sapphire substrate that portrayed the 2-theta reflection of phase analysis related to c-plane (0002) and (0004) at 34.59° and 72.79°, respectively, whereas the a-plane $(11\bar{2}0)$ and $(22\bar{4}0)$ sapphire substrate were at 37.85° and 80.61°, respectively [56]. Interestingly, Goel et al. reported the growth of $MoS_2$ on c-plane GaN that portrayed 2-theta reflections at 14.3°, 25.4°, 39°, and 58.4°, which corresponded to

multiple facets of MoS2 in (002), (004), (103), and (110), respectively [54]. The high intensity of c-plane GaN substrate for 2-theta reflection was observed at 34.5°. This in turn indicated the high crystallinity of the GaN substrate [54].

To further support the 2-theta phase-analysis measurement, on- and off-axis X-ray rocking (XRC) analyses have frequently been conducted to investigate the crystalline quality of the nominal growth planes [7,9,30,31]. The anisotropic characteristics and the planar defects of a semi-polar (11$\bar{2}$0) GaN epitaxial layer can be studied based on the reflection planes, as shown in Figure 6. In order to obtain these diffraction planes, the tilt with respect to the surface normal ($\chi$) and azimuth angle ($\Phi$) must be aligned.

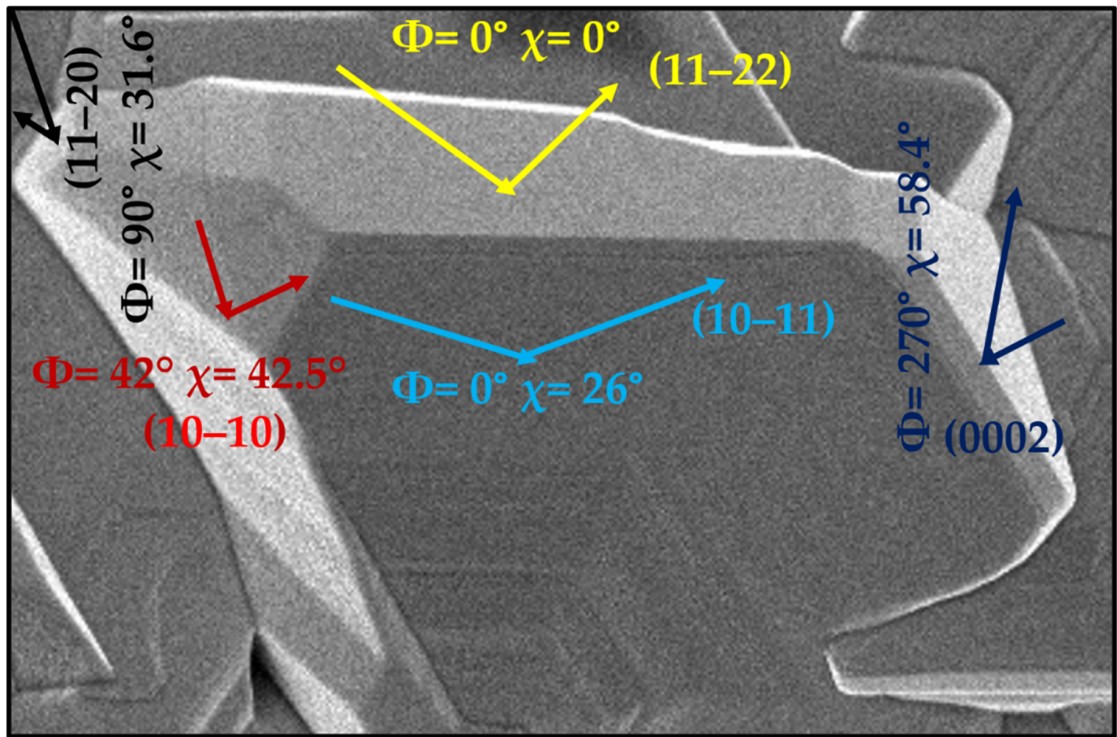

**Figure 6.** FESEM side-view image of GaN labelled with different diffraction planes for on- and off-axis XRC measurements.

It should be noted that the on-axis XRC would provide a single peak in an omega scan, in which this peak would exhibit different FWHM values at different phi angle scans, especially for non-polar GaN. This is mainly due to the anisotropic properties of non-polar GaN, which could lead to an orthorhombic distortion of the grown GaN [3,64]. In such a phenomenon, the on-axis XRC should be performed with at least two different angles that are parallel to (0002) and (10$\bar{1}$0) GaN planes to obtain a proper analysis of non-polar crystalline quality. Cai et al. reported a remarkably high quality of a-plane GaN grown on Si substrate by applying epitaxial overgrowth method with on-axis XRC FWHM of 325 and 380 arcsec along (0002) and (10$\bar{1}$0) GaN, respectively [9]. Another high-quality a-plane GaN epitaxial was reported by Yang et al., with FWHMs of 300 arcsec and 330 arcsec along (0002) and (10$\bar{1}$0) GaN, respectively [31]. Several reports on the growth of a-plane GaN on sapphire substrate resulted in a broadening of omega scan reflections with FWHM of ~1400 arcsec along (0002) GaN and ~2500 arcsec along (10$\bar{1}$0) GaN [7,9,17,23,30,64]. Additionally, the growth of nano-porous (0002) GaN on a-plane sapphire substrate using MBE by Aggrawal et al. portrayed an extra broadening in XRC FWHM compared to the typical grown c-plane GaN with an on-axis FWHM value of 2034 arcsec and 3445 arcsec along (002) and (102) GaN, respectively [56]. The broadenings of the on-axis XRC FWHMs mainly resulted from the nano-porous growth modes, which were mainly due to the interruption of the atom arrangements of the wurtzite crystal structure.

In addition to the aforementioned, it is worth mentioning that the off-axis XRC could provide crucial information on the defect density-related BSFs and of the non-polar GaN film [65–67]. However, only Pant et al. highlighted the importance of off-axis XRC measurement, and further interpretations of off-axis measurement were not available [23].

Another technique that provided vital information via X-ray diffraction was reciprocal space mapping (RSM). The RSM measurement could provide the accurate strain state of the epitaxial layers and reveal direct information on the lattice parameter [3,68,69]. Mukundan et al. included the RSM measurement along (11$\bar{2}$0) to study the strain state analyzing the tilting angle between GaN and sapphire peaks [3,30]. Pant et al. correlated the crystal quality of the non-polar GaN epitaxial lyres with the RSM measurements [23].

II.   Raman Spectroscopy

Room-temperature Raman spectroscopy measurement has been widely utilized to investigate the structural properties of GaN epitaxial layers [7,19,54,66]. The $E_2$ high mode wave vectors from Raman spectroscopy were very sensitive to stress, which could provide an advantage for determining the stress state of wurtzite GaN. The stress value of wurtzite GaN can be obtained by utilizing the following equation [70]:

$$\sigma xx = \Delta\omega/K \tag{4}$$

where $\Delta\omega$ is the difference in the $E_2$ high mode peaks between the grown GaN and perfect GaN, and K is the strain coefficient with a value of 4.2 cm$^{-1}$/GPa. The $E_2$ high peak for perfect GaN epitaxial layers is 567.6 cm$^{-1}$ [70,71].

The growth of a non-polar GaN epitaxial layer on sapphire substrate using MBE with and without plasma assistance reported by Gundimeda et al., Pant et al., Aggrawal et al., and Mukundan et al. illustrated $E_2$ high mode peaks at 567.84 cm$^{-1}$, 568.76 cm$^{-1}$, 566.61 cm$^{-1}$, and 569.25 cm$^{-1}$ with stress states of $-0.055$ GPa, $-0.123$ GPa, 0.230 GPa, and $-0.020$ GPa, respectively [7,19,30,56]. It is noted that the negative values indicated tensile states, whereas the positive values indicated compressive states of the stress. It is also interesting to observe the high value of the compressive stress state in the GaN MW structure as reported by Song et al. [57]. The $E_2$ high mode peaks portrayed a large shift towards the lower shift state compared to stress-free GaN. Meanwhile, the $MoS_2$ layer deposited on GaN substrate reported by Pant et al. exhibited two sharp peaks in Raman spectra at wave numbers of 383 cm$^{-1}$ and 407 cm$^{-1}$, which corresponded to the $MoS_2$ layer for in-plane $E_{2g}^1$ and out-of-plane $A_{1g}$ vibrational modes, respectively [54].

III.   Transmission Electron Microscopy

In order to further analyze the crystal quality and the surface morphology of the grown GaN thin films, TEM measurement is highly efficient due to the higher magnification scan size that enables a more detailed understanding of the growth mechanisms and defects as well as dislocations. Song et al. and Tsai et al. utilized TEM analysis in their reports, revealing the selective area electron diffraction (SAED) pattern. Their analyses portrayed high-quality and defect-free GaN micro- and nanowires [57,58]. Furthermore, a very interesting TEM analysis of a core-shell InGaN/GaN single-nanowire photodiode was reported by Zhang et al. [55]. The TEM analysis clearly proved the core-shell structure by performing a horizontal cross-section of a single GaN nanowire that comprised n-GaN as a core, surrounded by 30 pairs of InGaN/GaN quantum well and coated with p-GaN 175 nm thick. In addition to the aforementioned, Yang et al. utilized the TEM analysis to calculate the defect density within the non-polar GaN epitaxial layer as follows:

$$d = \frac{\beta}{x} \tag{5}$$

where x and $\beta$ represent the area size and the number of defects in the area, respectively [31]. A different approach reported by Mukundan et al. uses TEM analysis to obtain a clear insight into the condition at the interface of the GaN thin film and the substrate [30].

The sharp interface and the perfect atom arrangements at the interface suggest that their optimization successfully reduced the defect densities and obtained a high crystalline quality of a-plane GaN on r-plane sapphire substrate.

### 3.3.2. Morphological Analysis

The surface morphology and the related cross-sectional properties of the grown GaN epitaxial layers provide a significant advantage in understanding the principle of the efficiency of photodetection devices. These surface morphology analyses could be accomplished using either FESEM or SEM. The reports of the 2D growth of non-polar GaN thins films using MBE systems with and without plasma assistance resulted in a huge similarity in surface morphology by SEM study. These grown a-plane GaN films showed a typical a-plane GaN surface morphology with undulated and arrow-like structures, as reported by Gundimeda et al., Mukundan et al., Pant et al., and Mishra et al. [7,17,20,53]. It should be noted that the non-polar GaN growths encountered difficulty in obtaining flat and smooth surfaces, mainly due to the large asymmetric lattice mismatch between GaN film and the substrate, leading to in-plane strain distributions [65,67]. In such a surface condition, it is highly favorable to increase the surface-to-volume ratio, which would enhance the photo-detection system [7]. Interestingly, several researchers have proposed different methods to further enhance the surface-to-volume ratio by growing GaN with micro and/or nano structures such as MW, NW, NC, and NP GaN [32,56–58]. Hence, FESEM measurement would be very helpful to identify the structure of the GaN growth. Aggrawal et al. claimed that they obtained GaN in various structures from the FESEM images, including GaN thin film, NC, and NP [56]. In addition, it was further reported that the growth of nanowires was successfully achieved with a height over 1 μm and a width of 700 nm, 5 um, and 70 nm, as reported by Tsai et al., Zhang et al., and Patsha et al., respectively [32,55,58]. Additionally, Song et al. reported the growth of a-plane GaN MWs on patterned Si substrate, producing a length of several hundred μm with a thickness of 2 μm. A further cross-sectional FESEM analysis on the single GaN MW revealed a trapezoid shape with a mixture of semi- and non-polar facets [57].

Another a-plane GaN growth on Si substrate using the ELOG technique was reported by Cai et al. [9]. They revealed a smooth and abrupt surface with a stripe pattern of the ELOG mask that was obtained from FESEM surface analysis. The cross-sectional FESEM analysis illustrated the growth mechanism of a-plane GaN at the surface of the substrate, showing that the selective growth area contributed to the defect reduction by controlling the growth rate along the [0002] direction [9]. As reported by Goel et al., the growth of $MoS_2$ on GaN substrate portrayed flake-like structures that connected to each other, forming a thin layer [54]. In turn, it highly increased the surface ratio of the device, illustrating a similar principle as the micro- and nanostructure of GaN.

Additional characterization to further support the surface area analysis of FESEM is AFM. It should be noted that AFM analysis is one of the most appropriate measurements to provide vital information on the properties of surface topography, surface roughness, and defect density of the grown film. Goel et al. performed AFM analysis on the $MoS_2$ surface, revealing a smooth surface of the $MoS_2$ monolayer with a surface roughness of 0.3 nm [54]. On the other hand, Yang et al. utilized AFM measurement to analyze the BSFs and edge dislocation on a-plane GaN grown on sapphire substrate by etching the GaN film to reveal the defects on the surface [31]. Mukundan et al. and Pant et al. reported in their AFM measurements that the typical a-plane GaN topographies portrayed a wavy surface structure and the formation of pits with a surface roughness of between 1.5 and 4 nm [17,30]. In addition, an interesting finding by Mukundan et al. reported a correlation between the surface roughness and mobility towards the Ga adatoms present on the GaN surface [30].

### 3.3.3. Photoluminescence

Gundimeda et al. performed a room-temperature PL measurement that consisted of near band-edge (NBE), blue luminescence (BL), and yellow luminescence (YL) at 3.39 eV, 3.0 eV, and 2.2 eV, respectively. The low intensity of both BL and YL bands observed by Gundimeda et al. were correlated with the dissociation of excitons bound to neutral donors [7,71]. In contrast, Cai et al. reported a-plane GaN epitaxial layers grown on Si substrate that revealed a strong and narrow intensity of NBE peaks at 357.7 nm with a weak shoulder of a BL band; this phenomenon was related to the BSF densities [9,72]. Song et al. performed a temperature-dependent PL with a temperature range of 70 K to 290 K. The spectra illustrated a dominant peak of NBE for every temperature with a blue shift regarding the decreasing temperature due to the increase in the energy band gap. Furthermore, it was reported that the YL and BL bands did not result from the GaN microwires, promoting a proper crystal arrangement within the GaN microwires [57].

Apart from that, the room-temperature PL of a core-shell InGaN/GaN single NW was reported by Zhang et al. [55]. Their findings portrayed a dominant peak of NBE with a broadening shoulder of the BL band that resulted from the side facets of the quantum wells. The mentioned BL bands had a tendency to exhibit a red shift alternating the indium incorporation within the quantum well, which was mainly due to the dispersion of a self-assembled NW array [55]. Another analysis of GaN NW reported by Patsha et al. was performed under low-temperature conditions of 80 K. Their findings portrayed a dominant peak of NBE at 3.0 to 3.3 eV with a weak BL peak intensity at 3.4 eV [32]. In this study, the BL peaks were reported to be due to the transitions between shallow acceptor MgGa and the deep donors that resulted from the shifting of the BL bands at different Mg doping of the p-GaN [73]. The room-temperature PL spectra for the InGaN layer reported by Mukundan et al. revealed the high intensity of NBE emission with a slight blue shift with respect to the indium composition within the InGaN epitaxial layer [53]. The blue shift in the room-temperature PL was attributed to the changes in the thermal strain. In turn, the lattice properties and the atom arrangements at the interface of InGaN and substrate were changed [53]. Mukundan et al. further analyzed the optical properties by performing a temperature-dependent PL analysis of 10 K to 300 K. They reported that with the temperature use of 200 K, the PL spectra portrayed an obvious broadening of the NBE shoulder. This phenomenon might have resulted from the kinetics energy between electron–phonon interaction, resulting in a strong broadening of the PL bands [53,74].

The structural, morphological, and optical properties are well-established measurements that can be utilized to confirm the quality of the grown epitaxial layers. As for the structural properties, HR-XRD phase analysis was utilized by majority of the research groups within the selected article, whereas XRC and RSM analyses were the least to be reported. On- and off-XRC and RSM analyses were reported to investigate defects and/or dislocations and the strain/stress, respectively [2,3,31,66,75]. It should be noted that the highest crystal qualities as investigated via XRC analysis were 300 and 325 arcsec along [11$\bar{2}$0], as well as 330 and 380 along [10$\bar{1}$1], as reported by Yang et al. and Cai et al., respectively [9,31]. The lowest crystal qualities investigated via XRC were ~5250 and 2700 arcsec, as reported by Pant et al. and Aggarwal et al., respectively [23,56]. In contrast, based on the screened literature, only Pant et al. and Mukundan et al. reported RSM analysis to further investigate the crystal quality and the in-plane relaxation [23,30]. Mukundan et al. reported a slight enhancement in the relaxation of the epitaxial layer, which could lead to improved crystal quality. Pant et al. reported the least mosaicity and compositional inhomogeneity, which could have led to an enhancement in crystal quality. Hence, XRC and RSM measurements should be further utilized to confirm the structural properties, whereby by Williamson–Hall plot analysis could be obtained via the use of off-axis XRC [71,76,77].

In addition, it is worth mentioning that TEM analysis is a well-established analysis that can be utilized to confirm the densities of defects and/or dislocations [55,58,59]. Hence, further analysis by TEM is crucial to further support the quality of the as-grown GaN epitaxial layers. On the other hand, as can be discerned from Table 3, Raman spectroscopy analyses

were further utilized to confirm the crystal and/or structural quality [7,19,31,55,57,58]. Based on these studies, the lowest values of biaxial compressive stress were 0.05 and 0.07, as reported by Gundimeda et al. and Aggarwal et al., respectively [7,56].

As for surface morphology analysis, SEM measurements were reported by majority of the targeted articles, and a few research groups reported the use of the AFM analysis. It should be noted that the grain/terrace size distributions and the surface roughness of the non-polar GaN-based photodetector played a huge role in the performance of the device. In addition, it was reported that AFM analysis could be further utilized to confirm the defect densities [31,78]. In other words, the screw dislocation had an impact on the surface, whereby these dislocations could generate a tension tangential to the surface, leading to surface depressions that prevailed the surface [31,79,80]. In turn, there was a clear correlation between the AFM and the device performance. In addition, based on the screened articles, the lowest surface roughness value was reported to be <0.3 nm [55]. On the other hand, optical properties, namely, PL analysis, were reported by several research groups with respect to the targeted article. It also regarded a crucial category that could examine the quality of the epitaxial layer, which could have had a direct impact on the device performance.

## 4. Comprehensive Analysis of Bibliometric Analysis

It is noteworthy that the number of academic studies has been proliferating in non-polar GaN-based photodetectors. The focus on empirical contributions has led to an exponential increase in empirical studies in this research area. The rapid increase in such studies is a barrier to effectively summarizing and synthesizing the knowledge based on previous studies. To provide insight based on evidence from the previous studies, a transparent and systematic review performs a critical role in summarizing the findings of the previous literature. Various approaches have been utilized to reorganize the results of previous studies. However, bibliometric analysis could provide a transparent and systematic review based on scientific activity and statistical measurements [49,81,82]. In turn, it can achieve the analysis within more reliable objectives. The overwhelming quantity of information regarding non-polar GaN-based photodetectors is a challenge to reorganizing the information and summarizing the results of previous studies. Hence, the utilization of the R language for bibliometric analysis would be useful in reorganizing the results of such a changing field and conducting comprehensive analysis. Table 4 summarizes the results of previous studies. However, to determine the critical results of non-polar GaN-based photodetectors, the data visualization must be performed as simplified in the following sub-sections.

### 4.1. Annual Scientific Production

Figure 7 illustrates the number of publications covered by publication year from 2015 to 2021. The analysis shows that majority of the research articles were produced in 2018. These articles were in non-polar GaN due to the recent increase in issues associated with this matter. Using non-polar GaN would be vital for solving complex problems such as piezoelectric and spontaneous polarization. Hence, the literature on this scientific field has increased in recent years. As can be discerned, there was an increase of 20% in 2018 and 26% in 2020 and 2021. This growth was mainly due to the non-polar growth direction, providing solutions to overcoming piezoelectric and spontaneous polarization and photodetection application [9,17,83].

**Table 4.** Main information obtained from the selected articles.

| Description | Results |
|---|---|
| Timespan | 2015:2021 |
| Sources (journals, books, etc.) | 9 |
| Documents | 15 |
| Average years from publication | 3.13 |
| Average citations per document | 16.47 |
| Average citations per year per doc | 3.686 |
| References | 510 |
| Document types | |
| Research articles | 15 |
| Document contents | |
| Keywords plus (ID) | 169 |
| Author's keywords (DE) | 45 |
| Authors | |
| Authors | 83 |
| Author appearances | 113 |
| Authors of single-authored documents | 0 |
| Authors of multi-authored documents | 83 |
| Author collaboration | |
| Single-authored documents | 0 |
| Documents per author | 0.181 |
| Authors per document | 5.53 |
| Co-authors per document | 7.53 |
| Collaboration index | 5.53 |

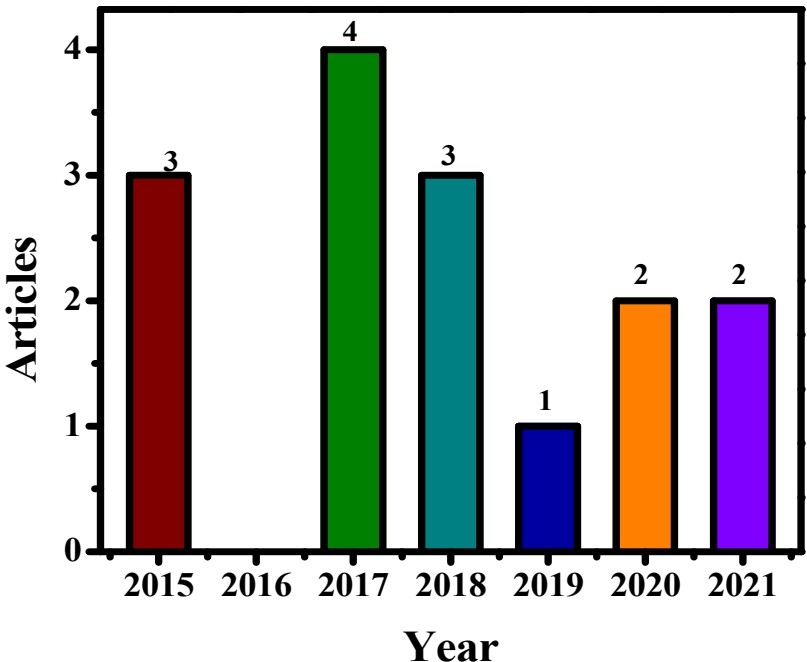

**Figure 7.** Publications per year of non-polar GaN-based photodetectors.

### 4.2. Word Cloud

A bibliometric keyword co-occurrence network analysis was performed as shown in Figure 8 with the help of R package [82,84]. The co-word analysis attempts to create a conceptual framework by mapping and grouping phrases that were extracted from the bibliographic information, such as keywords, titles, and abstracts. A word cloud can contribute to the creation of the study taxonomy. As shown in Figure 8, the most relevant sections of the textual data were arranged. This method will assist academics and practitioners in comparing the various components in order to identify commonalities and variations between the larger and bolder words. Additionally, word cloud analysis can enable academics and industries to recognize the advantages of non-polar GaN-based photodetectors by clustering words of varying sizes.

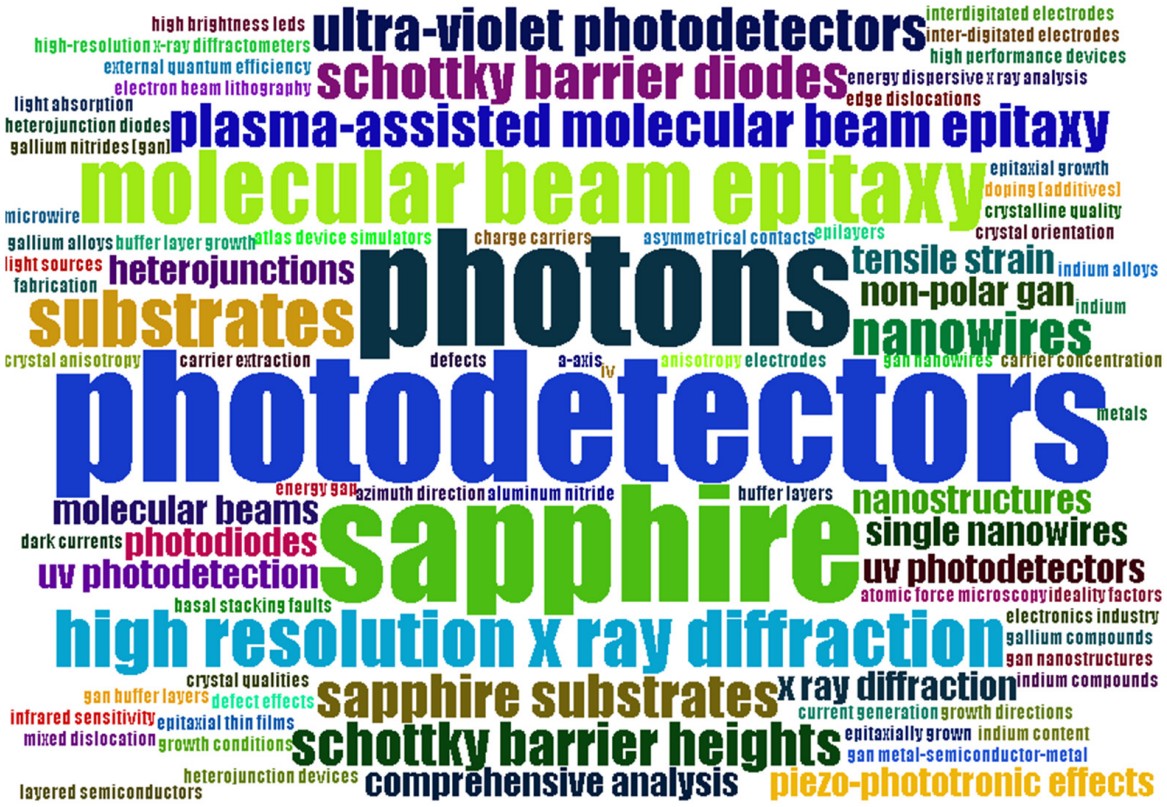

**Figure 8.** Keyword plus co-occurrence network of non-polar GaN-based photodetectors.

### 4.3. Factor Analysis

The similarity function might cluster data on co-citation, bibliographic coupling, and co-occurrence based on the factorial analysis. As shown in Figure 9, there are many advantages to using the similarity function to calculate the Jaccard coefficient and inclusion index. The factorial analysis of non-polar GaN-based photodetectors is shown in Figure 9. Based on the similarity index, the keywords for non-polar GaN-based photodetectors were classified into two clusters. This function was widely used to mine the literature for pertinent information, and it could also be employed for the topic of non-polar GaN-based photodetectors [49,81]. The similarity index was deemed to be high since the problems addressed in the reviewed literature were comparable. It can also be observed that the similarity function generated homogenous groups of linked research, assisting academics and beneficiaries in differentiating and recapitulating the pertinent information.

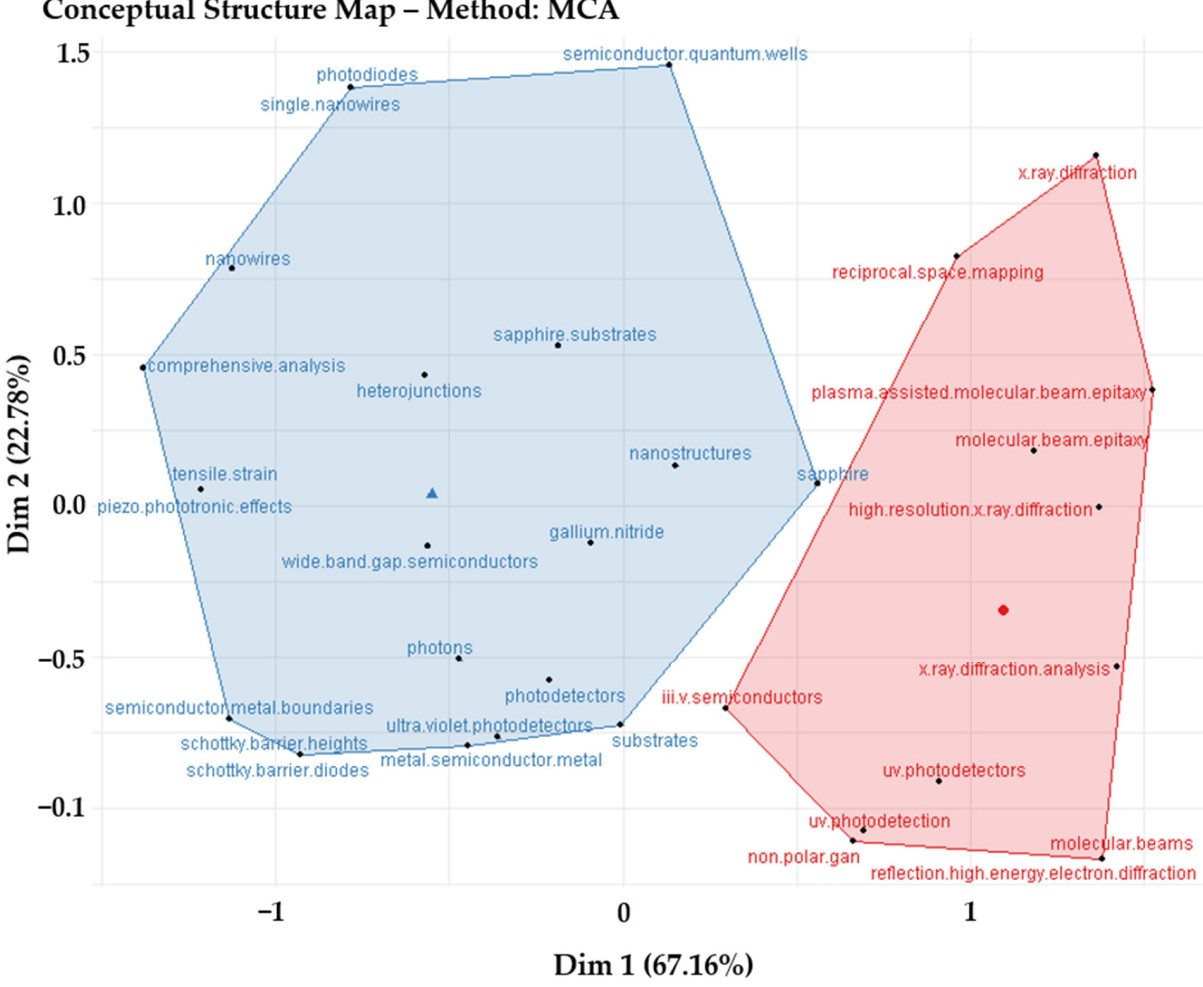

**Figure 9.** Conceptual map and keyword clusters of non-polar GaN-based photodetectors.

## 5. Challenges, Motivations, and Future Opportunities

### 5.1. Challenges

It should be noted that performing III-V devices, especially non-polar III-nitride semiconductors, has suffered from extended defects and dislocations presented within the microstructures of the layers. This is mainly due to lattice mismatch regarding the substrates, although these devices have a relative advancement in 2D technology in terms of UV photodetection [19,31,55]. In addition, the mechanism of the influence of the defects and dislocations and their effects towards the performance of the device is still unclear [31]. The lattice mismatch between the substrate and the grown thin films could play a huge role in deciding on the quality of GaN-based optoelectronic devices [23]. It is noteworthy that non-polar GaN homoepitaxial growth is ideal in terms of the enhancement of crystal quality and lattice match. However, the lack of large-sized GaN single-crystal substrate is a challenging factor in the development of GaN-based photodetectors [9,57]. In turn, it has led to the use of GaN epitaxial layers grown on large-sized substrates such as Si and sapphire substrates [56]. However, upon using these two candidates to grow non-polar GaN-based optoelectronics, huge challenges have been encountered owing to lattice mismatch between Si or sapphire and GaN epitaxial layers, leading to the formation of defects and densities [9,17,85]. It was reported by Yang et al. that the densities of BSFs, edge dislocation, and screw or mixed dislocation could influence the response time, responsivity, and dark current of non-polar GaN-based photodetectors [31]; BSF and edge dislocation would reduce the responsivity due to the decrement of the electron mobility, whereas the dark current was increased due to the screw or mixed dislocation. The response time would

be increased owing to these defects and dislocations via the formation of traps to recombine the holes with electrons, resulting in a huge delay for the carriers to escape. It should also be noted that the situation became worse for Si as compared to sapphire substrate. The growth of non-polar GaN on Si substrate has become difficult due to the epitaxial relationship in compatibility between Si substrate and non-polar GaN irrespective of any orientation [85]. The growth of GaN epitaxial layers on Si substrates has suffered from major challenges such as cracking and Ga meltback etching [9]. For the meltback etching issue in the polar GaN using (111) Si substrate, the growth of a thick AlN buffer layer could circumvent this challenge, thus eliminating the Ga meltback etching. The growth of non-polar GaN must be obtained on inclined {111}, and the formation of Ga meltback etching is tremendously increased. In addition, the large mismatch between the Si substrate and the GaN could lead to severe cracking. Thus far, the growth of non-polar GaN could mainly be achieved using pattern Si substrate via a selective growth method [9,72,85,86].

Additionally, the growth of a GaN epitaxial nanostructure was reported on $(11\bar{2}0)$ sapphire substrates via LMBE [56]. They reported that the nitridation treatment and the growth of buffer layer GaN at a low temperature could critically affect the structural properties and the surface morphology. They also reported a limitation under the high nitrogen-rich level, whereby the diffusion length for the laser-ablated Ga and the adatom of the $GaN_{1-x}$ growth was bridled due to the nitrogen species plasma, leading to the formation of 3D GaN growth. Another challenge that was encountered during the growth of an ensemble and single nanowire was reported by Patasha et al. [32]. They reported the generation of the defect states that resulted from the inhomogeneity in Mg dopants of an ensemble nanowire device that was mainly due to the factor of high ideality. It should be highlighted that these heterojunctions have suggested exceptional enhancement of light-harvesting devices and high-speed electronics. However, the precision in controlling the doping to produce large-area heterostructures with proper and abrupt interface has remained challenging [87]. Another challenge of the solid-state miscibility gap between indium nitride (InN) and GaN is to grow $In_xGa_{1-x}N$ alloy-based photodetector devices owing to the vast difference in the interatomic spacing [54]. In contrast, it was reported that the photodiode frequency response is faster than photoconductive NW devices, which might be attributed to the defect densities and trapping effects on the surface [55].

In contrast, to tune and/or enhance the performance of the electronic materials in conjunction with a piezoelectric semiconductor, the piezotronic effect has arisen as a new means of doing so [57,58]. This could be obtained via the use of the piezoelectric potential (piezopotential), which was generated in response to the external stress. However, the piezopotential generated in the piezoelectric semiconductor would be critical to piezo-phototronics and piezotronics [58]. In addition, it should be noted that there were no adequate studies of piezo-phototronic and piezotronic effects towards the modification of organic/inorganic hybrid heterojunctions/interfaces within a-axis GaN [57]. It would extend the development of applications with the use of III-V semiconductors in optoelectronics and/or electromechanics [57].

There are several factors that formed enormous challenges with the working principle of photodetectors [17]. The device and the fabrication material could play a huge role in determining the adaptable usages in hard conditions. Another factor that could affect the efficiency of the device is self-powered detection. Several research groups proposed different methods for self-powered photodetection via different configurations, namely, hybrid materials in conjunction with nanomaterial combinations, 2D materials as electrodes, and the asymmetric IDEs that have different materials for the electrodes [17,39,88–94]. The final challenging aspect reported was the selective wavelength, in which it could reduce the background noise generated by the spectrum [17]. It is noteworthy that optical filters were used to eliminate the visible and infrared noise to obtain a selective UV detection [17]. However, the aforementioned method could lead to an increment in the cost and complexity. Additionally, the lack of self-powered non-polar GaN-based UV photodetectors has led to a significant challenge. These self-powered UV photodetectors have been achieved with other

materials and/or hybrid structures that could affect the properties of UV photodetection, such as robustness, radiation hardness, and chemical inertness [17]. Another limitation is the forward bias current, which was limited to a maximum value to circumvent accidental breakdown of the device. This might have resulted from spurious electrical spikes and/or joule heating [32].

*5.2. Motivations*

The development of UV photodetectors could pave the way for a wide range of biological, industrial, and environmental applications, such as space communications, optical communications, atmospheric ozone detection, environmental monitoring, bio-photonics, missile detection, flame sensors, and so forth [7,9,17,30]. Given that wide and direct bandgap materials such as GaN (3.43 eV) are promising candidates in UV detection, it is expected to exhibit superior performance compared to Si-based UV photodetectors due to the indirect bandgap of Si (1.1 eV) [9,23,53,56]. GaN-based materials have received great attention because of their superior properties, such as high thermal stability and high radiation resistance. In turn, it can lead to promising possibilities for high performance (high gain and high speed) [7]. It should be noted that crystallographic orientation and surface polarity would play a huge role in determining the performance of the photodetector devices [9,95]. Currently, the effect of spontaneous and piezoelectric polarization leads to internal electric fields that negatively affect the performance of GaN-based photodetectors in terms of response speed and efficiency [7,9,19,55]. However, the growth of GaN-based photodetectors along the non-polar direction promoted a zero-polarization field. In turn, it could portray higher efficiency and faster response compared to their c-oriented counterparts, although the crystal quality of non-polar epitaxial layers has been a huge challenge [7,19,56]. Therefore, optimizing the growth conditions of non-polar GaN epitaxial could lead to a benchmark without compromising the properties of photodetection [19]. Additionally, the anisotropic properties of non-polar GaN were found to be promising for the fabrication of polarized UV photodetectors in the field of remote sensing and airborne astronomical navigation [31]. Further motivation was seen with the use of Si substrate, as reported by Cai et al. [9]. They reported that the growth of GaN-based photodetectors has drawn increased attention due to the huge demand to integrate the III-nitride semiconductors with silicon technologies [9].

It was reported that there was great potential for the use of InGaN alloy with a relatively higher indium composition to investigate its effect on the optical bandgap [53]. Another motivation was the use of 2D InGaN/GaN heterostructures due to their merits in photodetection [55]. A report of NW GaN-based photodetectors suggested a significant improvement in the use of additional surface treatment [58]. Apart from that, it was seen that the use of single-nanowire p-GaN/n-Si devices portrayed higher efficiency in the electrical elements of the photodiode characteristics compared to the ensemble nanowire device [32]. In contrast, it was reported that the growth of 2D layered transition metal dichalcogenide materials on 3D semiconductor substrates could overcome the critical challenges encountered by the new generation of optoelectronics due to their optical and electronic properties [54]. Nanoscale organic/inorganic hybrid heterostructures were another motivation, owing to the simplicity of fabrication along with the promising optical and electrical properties of inorganic semiconductors and the flexibility of organic components [57].

*5.3. Future Opportunities*

5.3.1. Growth Conditions

Future research work on the growth of non-polar III-V semiconductor material-based optoelectronics is discussed in this section. It should be noted that the use of nanowire devices portrayed promising performance of non-polar GaN-based photodetectors. However, the use of a multi-quantum well (MQW) is lacking, although the phonon scattering from the carrier extraction obtained from the quantum well (QW) is crucial. The cluster

of InGaN can be further optimized in terms of cluster size and indium content to obtain quantum confinement. In addition, other techniques to further reduce defects and dislocation with the microstructure of non-polar GaN should be considered, such as pulse techniques, multilayer strained layers, strained layer superlattices (SLSs), ammonia ($NH_3$) treatment, etc. Moreover, further optimization of the 2D growth of layered transition metal dichalcogenide materials on 3D semiconductor materials can be conducted due to their optical and electronic properties.

Besides, conducting research on III-V on graphene and flexible graphite is crucial to further developing a technology that is independent of sapphire and/or Si substrates. In addition, studies of NW p-GaN/n-Si are lacking, in which the doping variants of p-type GaN could contribute to the body knowledge of this field in terms of photoresponsivity and the response speed. In addition, other in-situ methods to optimize the growth of the non-polar GaN templates could be used to further improve the performance of the photodetectors, such as $NH_3$ flow, disilane or silane flow, or biscyclopentadienyl magnesium ($Cp_2Mg$) flow. These methods can be utilized to reduce defects and/or dislocation, leading to significant improvements in the device performance.

Nanoscale organic/inorganic hybrid heterostructures portrayed an immense improvement in the optical and electrical properties. Thus, further enhancement in the structural properties would be essential to achieve a benchmark of hybrid photodetectors. These structures would have the potential to enhance the device performance in terms of photon detection and emission. In addition, it was seen that the use of an $MoS_2$ heterojunction could enhance the performance of the photodetector. Hence, the use of an $MoS_2$ heterojunction on a GaN epitaxial layer is yet to be studied in detail. Furthermore, integrating non-polar GaN (TF, NW, and MW) with graphene would be a promising approach owing to their compatibilities in terms of optical and electrical properties, in which the photoresponsivity and the response speed can be considerably enhanced.

### 5.3.2. Fabrication

Schottky barriers of Au/GaN were used to obtain self-powered photodetection and further improve the photodetection at high applied bias because of the difference in the Schottky barrier height [17]. The difference in the height could generate an electrical field towards the higher Schottky barrier [17]. Hence, other types of metal contacts should be further used to investigate the above phenomenon. In terms of non-polar semiconductor materials, the photoresponsivity and response time could be highly dependable on the azimuth angle, especially for in-plane crystal anisotropy [23] Thus, further optimization should be accomplished within the in-plane crystal anisotropy.

### 5.3.3. Characterization

As for the structural properties, further analysis on off-axis XRC should be implemented to further investigate dislocation and defect increase/decrease and their correlation with the growth conditions. In addition, Williamson–Hall plot analysis can be achieved with the use of off-axis XRC along [10$\bar{1}$0] and [20$\bar{2}$0] [65,76,77]. Based on the screened articles, a few studies reported the use of off-axis XRC analysis. Therefore, there is a huge demand for such analyses to be implemented to further examine the microstructural properties. On the other hand, based on the screened articles of non-polar GaN-based photodetectors, only two research groups reported the use of RSM analysis to investigate the crystal quality and the in-plane relaxation [23,30]. Thus, RSM analysis can mainly contribute to the in-plane relaxation and its correlation to the prismatic staking faults (PSFs) [64,75,96]. The tilt of the materials' peaks with respect to the substrate's peak and the existing streak can be further investigated. In turn, a correlation between the in-plane relaxation and the device performance be systematically studied. Hence, XRC and RSM measurements should be further utilized to confirm the structural properties.

Further analysis of grain size distributions on a nano-scale level using AFM can be implemented to investigate the morphological properties of non-polar GaN-based

photodetectors. It was previously reported that there would be a correlation between the closely packed/loosely packed grains and the decrease/increase in the defects within the microstructure of the GaN epitaxial layers [1,36,97–100]. Furthermore, our group reported the use of statistical analysis (histogram analysis) of the grain size distributions by AFM measurements [100,101]. It was found that these analyses were promising to examine the morphological property-induced crystal quality. However, these investigations were performed along the semipolar orientation. Hence, these investigations should be further implemented to further support the material quality and its correlation to the device performance. In other words, the screw dislocation and/or defects would have an impact on the surface, whereby these dislocations/defects could generate a tension tangential to the surface, leading to surface depressions that were prevailing on the surface [31,79,80,102].

## 6. Conclusions

In conclusion, recent studies of non-polar GaN-based photodetectors over a six-year period (January 2015–August 2021) were systematically reviewed. Several approaches, techniques, and/or methods as well as properties that were reported by previous studies were taxonomized. The systematic classifications, bibliometric analysis, challenges, motivations, and future opportunities of non-polar GaN-based photodetectors were also discussed. The reviewed articles were taxonomized into three categories. The first category included the growth and fabrication, whereas the second category comprised the electrical properties, followed by the third category of the structural, morphological, and optical properties of non-polar GaN-based photodetectors. In addition, a bibliometric analysis was utilized to provide a transparent and systematic review that portrayed a critical analysis to summarize the findings non-polar GaN-based photodetectors. The future opportunities of non-polar GaN-based photodetectors were highlighted to provide future perspectives in terms of their growth conditions, fabrication, and characterizations. In turn, several research opportunities are available to further improve the technology of non-polar GaN-based photodetectors Hence, it is deduced that these contributions would enable better understanding of non-polar GaN-based photodetectors for future endeavors.

**Author Contributions:** Conceptualization, O.A.-Z.; supervision, A.S. and N.N.; funding acquisition, A.A.; methodology, A.K.; writing—original draft, O.A.; data curation, M.G.; validation, E.T.A.; investigation, Y.Z.; review and editing. All authors have read and agreed to the published version of the manuscript.

**Funding:** We acknowledge the Strategic Research Fund (SRF) by the Malaysian Ministry of Science, Technology and Innovation (MOSTI) MIMOS/CEO/2021/10(094), Collaborative Research in Engineering, Science & Technology (CREST), Malaysia for (PV007-2019)(T13C2-17) and Collaborative Research in Engineering, Science and Technology R&D Grant No. A154 (UTHM) & P28C1-17(CREST).

**Institutional Review Board Statement:** Not applicable.

**Informed Consent Statement:** Not applicable.

**Data Availability Statement:** Not applicable.

**Acknowledgments:** The authors would like to thank Alhamzah Alnoor for his help in analyzing the bibliometrix.

**Conflicts of Interest:** The authors declare no conflict of interest.

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
