# Peer review of "Non-Polar Gallium Nitride for Photodetection Applications: A Systematic Review"

_coatings, doi:10.3390/coatings12020275_

Round 1

Reviewer 1 Report

This paper systematically reviews the research on non-polar GaN photodetectors. The article is rich in materials and relatively detailed. However, the overall content is somewhat lengthy, and the introduction of basic knowledge accounts for a large proportion. From the perspective of the first half, it is not so much an article as an introduction part of a graduation thesis. A good review requires the author to have a strong ability to extract and refine information first, and then be good at summarizing. Finally, if you can express some of your own views, the article will be sublimated again. Considering the workload of this article, the reviewers make several suggestions:

  1. The author should consider whether the second part of the article is redundant;
  2. The pictures in the article should preferably be the experimental data of the cited literature, or compare the processed data of multiple articles, which will be more convincing;
  3. Refine the language and clarify the conclusions drawn from the summary;
  4. Pay more attention to formatting, such as capitalization and subscripts.

Author Response

Response to the Editor and Reviewer’s comments

Coatings

“Non-polar gallium nitride for photodetection applications: A systematic review”

Dear Editor and Reviewers,

Thank you for your useful comments and suggestion on the structure and scientific content of our manuscript. We have modified the manuscript accordingly, and detailed correction listed below point by point:

# Reviewer 1

This paper systematically reviews the research on non-polar GaN photodetectors. The article is rich in materials and relatively detailed. However, the overall content is somewhat lengthy, and the introduction of basic knowledge accounts for a large proportion. From the perspective of the first half, it is not so much an article as an introduction part of a graduation thesis. A good review requires the author to have a strong ability to extract and refine information first, and then be good at summarizing. Finally, if you can express some of your own views, the article will be sublimated again. Considering the workload of this article, the reviewers make several suggestions:

  1. The author should consider whether the second part of the article is redundant;

Action/amendment: Thank you for the highlighted comment. The second part of the manuscript (Systematic Literature Review Protocol) is one of the the main stars of the manuscript as it highlights the search process, inclusion and exclusion criteria, data extraction, study selection and the strategy. In turn, the readers can understand why these articles were included within the scope of our manuscript. Therefore, it was an essential part to include within the study of non-polar GaN-based photodetector.

  1. The pictures in the article should preferably be the experimental data of the cited literature, or compare the processed data of multiple articles, which will be more convincing;

Action/amendment: Thank you for the highlighted comment. The experimental data of the screened articles were included in the first taxonomy (Growth and Fabrication), which surrounds the experimental procedures. Additionally, the comparisons of the proposed categories in growth and fabrication were included in pages 16, line 3-26. Furthermore, comparisons in terms of the electrical properties were included in page 25, line 4-16. In contrast, a comparison of the structural morphological and optical properties based on the targeted articles were included in page 30 (line 7-16, (line 22-26), (31-37). Finally, for future opportunities were discussed in terms of growth condition, fabrication and characterization in 37, and 38.

  1. Refine the language and clarify the conclusions drawn from the summary;

Action/amendment: Thank you for the highlighted comment. The language of the summery was refined to clarify the conclusion of the manuscript as shown in 37.

  1. Pay more attention to formatting, such as capitalization and subscripts.

Action/amendment: Thank you for the highlighted comment. We are sorry for the unintentional mistakes of the formatting. The comment has been addressed accordingly, and the capitalization and subscripts were corrected.

Reviewer 2 Report

The authors did a good job putting together a comprehensive review article on Nonpolar GaN for photodetector applications. The methodology of screening and choosing the literature outlined in the review article is especially important and useful for researchers interested in undertaking systematic review of related material. The results of the systematic review have been comprehensively written and the structure of the paper is sound and well organized. I recommend that this review article be published in Coatings pending requirements set by the other reviewer as well as the editor. The paper needs a minor run through of language usage i.e. Spelling and grammar check.

Author Response

Response to the Editor and Reviewers’ comments

Coatings

“Non-polar gallium nitride for photodetection applications: A systematic review”

Dear Editor and Reviewers,

Thank you for your useful comments and suggestion on the structure and scientific content of our manuscript. We have modified the manuscript accordingly, and detailed correction listed below point by point:

# Reviewer 2

The authors did a good job putting together a comprehensive review article on Nonpolar GaN for photodetector applications. The methodology of screening and choosing the literature outlined in the review article is especially important and useful for researchers interested in undertaking systematic review of related material. The results of the systematic review have been comprehensively written and the structure of the paper is sound and well organized. I recommend that this review article be published in Coatings pending requirements set by the other reviewer as well as the editor. The paper needs a minor run through of language usage i.e. Spelling and grammar check.

Action/Amendment: Thank you for the comment. The comment has been addressed and the language usage, spelling and grammar mistakes have been corrected throughout the whole manuscript.

Reviewer 3 Report

The review article is well structured and comprehensive however, there are several things missing that could improve the article before it can be considered for publication: 

(1) Table 1 needs column partitions to separate the categories “Substrate,” “Phtodetector Geometry,” “Contacts,” “Layer Geometry”, “Technique,” and “Ref” to visually aid the reader.  Currently the table columns bleed onto one another.

(2) Figure 3 does not show the different structures of metal contacts and device geometry. It is only referencing one type.  As the authors discuss in Section 3.1.2 there are the four types of geometry MSM, pn, pin, and hybrid.  These four types must be put in Fig. 3.  These figures can be taken from the references and cited or the authors can make their own figures.

(3) The authors need to properly summarize the results of their review. Merely paraphrasing the results without commentary is not the essence of a review article. A review article must have some recommendations and thorough comparisons between references so that the reader can be guided in making an informed decision on the best techniques in the field. At the end of each subsection, the authors must make a short summary and commentary/recommendation/opinion about the subsection references that have been reviewed.

(4) The authors should include selected important data, images(SEM etc.) and graphs from the references that are considered a highlight or featured work from each subsection. 

(5) The review article needs some moderate work on improving the English usage and style (grammar and spelling).

Author Response

Response to the Editor and Reviewers’ comments

Coatings

“Non-polar gallium nitride for photodetection applications: A systematic review”

Dear Editor and Reviewers,

Thank you for your useful comments and suggestion on the structure and scientific content of our manuscript. We have modified the manuscript accordingly, and detailed correction listed below point by point:

# Reviewer 3

The review article is well structured and comprehensive however, there are several things missing that could improve the article before it can be considered for publication: 

  • Table 1 needs column partitions to separate the categories “Substrate,” “Phtodetector Geometry,” “Contacts,” “Layer Geometry”, “Technique,” and “Ref” to visually aid the reader.  Currently the table columns bleed onto one another.

Action/amendment: Thank you for the highlighted comment. Sorry for the unintentional mistake. The comment has been addressed and the partitions to separate the categories “Substrate,” “Photodetector Geometry,” “Contacts,” “Layer Geometry”, “Technique,” and “Ref” have been added to Table 1 in page

  • Figure 3 does not show the different structures of metal contacts and device geometry. It is only referencing one type.  As the authors discuss in Section 3.1.2 there are the four types of geometry MSM, pn, pin, and hybrid.  These four types must be put in Fig. 3.  These figures can be taken from the references and cited or the authors can make their own figures.

Action/amendment: Thank you for the highlighted comment. The comment has been addressed and the four types of geometry MSM, PN, P-i-N, and hybrid structures have been included in Figure 4.

  • The authors need to properly summarize the results of their review. Merely paraphrasing the results without commentary is not the essence of a review article. A review article must have some recommendations and thorough comparisons between references so that the reader can be guided in making an informed decision on the best techniques in the field. At the end of each subsection, the authors must make a short summary and commentary/recommendation/opinion about the subsection references that have been reviewed.

Action/amendment: Thank you for the highlighted comment. The comment has been addressed and a short summary about each category in pages 16, (line 3-26), page 25 (line 4-16) and page 30 (line 7-16), (line 22-26), (31-37). Finally, for future opportunities were discussed in terms of growth condition, fabrication and characterization in 37, and 38.

  • The authors should include selected important data, images (SEM etc.) and graphs from the references that are considered a highlight or featured work from each subsection.

Action/amendment: Thank you for the highlighted comment. Due to the short time of the review, it is not possible to obtain the approval from journal regarding SEM images and the graphs. However, we have included FESEM side-view image of GaN labelled with different diffraction planes for on- and off-axis XRC measurements in page 25 and 26 (line 313-319). In turn, it would elaborate about the diffraction planes, the tilt with respect to the surface normal (χ) and azimuth angle Φ. We sincerely apologize and we hope that the Editor and Reviewer can understand the current situation.

  • The review article needs some moderate work on improving the English usage and style (grammar and spelling).

Action/amendment: Thank you for the highlighted comment. We are sorry for the unintentional mistakes of the formatting. The comment has been addressed accordingly and the language of the manuscript was improved, including grammar and spelling.

Reviewer 4 Report

Non-polar gallium nitride for photodetection applications: A systematic review is very interesting review paper, but some minor improvement is required.

Line 29, line 30: Finally, challenges, motivation, and future opportunities of non-polar gallium nitride-based photodetectors are presented. Can you give an information about synthesis of these powders.

Line 70:  [2,7,19,23,29,33-35]. . This (please to delete ".")

 Line 120, 121: the date of the extraction process (for GaN), and the data synthesis of ....(GaN). Please to explain it better!

 Line 175: provide a new direction for future research (which subject: application of powders? Synthesis of powder? recycling of powder?)

Line 192: Figure 1. Flowchart of search query and inclusion criteria for the systematic review protocol (please to add: keywords for this search)

After Line 206 (Page 6) no information about Lines. Why? it is difficult to follow text.

page 10: . The(delete .)

Page 13: It is noteworthy that with the ability to produce a mass scale up production (Kg/day? tons?)

 Page 16, Line 38: It should be highlighted that non-polar nanostructure portrayed lower quality as compared with conventional polar nanostructured ..(materials? powders?)

Page 30, line 488: 4. Comprehensive analysis of bibliometric (data?). Is this title complete?

Author Response

Response to the Editor and Reviewers’ comments

Coatings

“Non-polar gallium nitride for photodetection applications: A systematic review”

Dear Editor and Reviewers,

Thank you for your useful comments and suggestion on the structure and scientific content of our manuscript. We have modified the manuscript accordingly, and detailed correction listed below point by point:

# Reviewer 4

Non-polar gallium nitride for photodetection applications: A systematic review is very interesting review paper, but some minor improvement is required.

  1. Line 29, line 30: Finally, challenges, motivation, and future opportunities of non-polar gallium nitride-based photodetectors are presented. Can you give an information about synthesis of these powders?

Action/amendment: Thank you for the highlighted comment. The comment has been addresses and answered within the abstract in line 30-32. In addition, as highlighted in the last section of the manuscript (5.3), the future opportunities of non-polar gallium nitride-based photodetectors in terms of growth condition, fabrication and characterization are highlighted. In contrast, the information of about the synthesis of the materials is highlighted in subsection 3.1 (growth and fabrication), surrounding substrate use, photodetector geometry, contacts, layer geometry, and technique.

  1. Line 70: [2,7,19,23,29,33-35]. . This (please to delete ".")

Action/amendment: Thank you for the highlighted comment. Sorry for the mistake. The comment has been addressed accordingly in line 74.

  1. Line 120, 121: the date of the extraction process (for GaN), and the data synthesis of ....(GaN). Please to explain it better!

Action/amendment: Thank you for the highlighted comment. Sorry for the mistake. The comment has been addressed accordingly in page 3, line 125-131.

  1. Line 175: provide a new direction for future research (which subject: application of powders? Synthesis of powder? recycling of powder?)

Action/amendment: Thank you for the highlighted comment. The comment has been addressed and highlighted in line 188-190. In addition, the future opportunities in terms of growth conditions, fabrication and characterization are highlighted in page 37-38.

  1. Line 192: Figure 1. Flowchart of search query and inclusion criteria for the systematic review protocol (please to add: keywords for this search)

Action/amendment: Thank you for the highlighted comment. The comment has been addressed and the keywords for this search were included within the caption of Figure 1.

  1. After Line 206 (Page 6) no information about Lines. Why? it is difficult to follow text.

Action/amendment: Sorry for the inconvenience caused. After we added a break to the word sequence to change the page to landscape, the numberings of the lines were gone. It was essential for Figure 2 to change it to landscape. 

  1. page 10: . The(delete .)

Action/amendment: Sorry for the unintentional mistake. The comment has been addressed in page 11, line 11.

  1. Page 13: It is noteworthy that with the ability to produce a mass scale up production (Kg/day? tons?)

Action/amendment: Thank you for the comment. The comment has been addressed the sentence changed to be more specific as shown in page 13, line 49-50.

  1. Page 16, Line 38: It should be highlighted that non-polar nanostructure portrayed lower quality as compared with conventional polar nanostructured ..(materials? powders?)

Action/amendment: Thank you for the comment. The comment has been addressed the sentence changed to be clear as shown in page 17, line 36.

  1. Page 30, line 488: 4. Comprehensive analysis of bibliometric (data?). Is this title complete?

Action/amendment: Thank you for the comment. The comment has been addressed in page 31, line 532.

Round 2

Reviewer 1 Report

The authors have replied to our comments and have made the changes in the article. Now I recommend this paper to be published.

This manuscript is a resubmission of an earlier submission. The following is a list of the peer review reports and author responses from that submission.

Round 1

Reviewer 1 Report

In this work, studies about non-polar GaN based photodetector over a 7-year period were systematically reviewed. The reviewer found that this work will be of interest to other researchers in scientific and engineering community of III-V family compound semiconductors. In the view of my point, before the consideration for publication, the following issues needs to be addressed:

1) The introduction have room to be further improved. The authors write: “Over the past decade, III-V semiconductor materials have showed promising outcomes for the applications of optoelectronic devices, such as light-emitting diodes (LEDs), laser diodes (LDs) and photodetectors [1-7].” The general reference list seems a bit thin, considering the evolution in the field within the recent years. Several recent references concerning on GaN-based LEDs, such as Optics Express 27, A669 (2019); Applied Physics Letters 118, 182102 (2021); Nano Energy 2020, 69, 104427, etc., should be added, so that the readers can be clear about the state-of-the-art of this topic.

2) The authors argued: “Different substrates were utilized based on the obtained articles, namely c-plane sapphire, r-plane sapphire, a-plane sapphire substrate and Si substrates as shown in Figure 3.” Can the authors include SiC substrate and GaN bulk substrate?

3) The author should provide schematic device structure of GaN-based UV detector in the manuscript. This will make this paper more readable.

4) What is a-GaN? Do the authors mean a-plane GaN?

5) The authors write: “An ultrahigh responsivity.” This sentence is incomplete.

6) The authors wirte: “The growth of non-polar GaN on plana Si substrate” What is the meaning of plana Si substrate?

7) “semiconductor substrates can be conducted due to their due to their optical and electronic properties” should be revised to be “semiconductor substrates can be conducted due to their optical and electronic properties”.

8) “Hence, other types of metal should be further used the investigate the above phenomenon.” should be revised to be “Hence, other types of metal should be further used to investigate the above phenomenon.”

9) There are some grammatical errors in the manuscript, although most of them do not obscure the understanding of the technical contents. However, I believe that the paper should be proof-read for English before it is submitted. For example:

--“III-V semiconductor materials have showed promising…” should be corrected to be “III-V semiconductor materials have shown promising…”

--“The exciton binding energy of these materials are (40-52 meV)…” should be corrected to be “The exciton binding energies of these materials are (40-52 meV)…”

--“…impairing devices’ efficiency such response of rise and fall time…” should be corrected to be “…impairing devices’ efficiency such as response of rise and fall time…”

--“…approach that can be utilize to determine a definitive topic in the literature…” should be corrected to be “…approach that can be utilized to determine a definitive topic in the literature…”

--“…conducted on the on the five major databases within 6 years” should be corrected to be “…conducted on the five major databases within 6 years”

--“It should be noted that the Boolean operators used here owing to their data  source and access capabilities” should be corrected to be “It should be noted that the Boolean operators were used here owing to their data source and access capabilities”

--“More interestingly, the behavior of a-GaN epitaxial layer towards the extended defects were investigate” should be corrected to be “More interestingly, the behavior of a-GaN epitaxial layer towards the extended defects were investigated”

--“The work comprise non-polar a-plane GaN epitaxial films were grown on an r-plane sapphire” should be corrected to be “The work comprise non-polar a-plane GaN epitaxial films grown on an r-plane sapphire”

--“It is important to studies the effect…” should be corrected to be “It is important to study the effect…”

--“due to of ultra-high vacuum (UHV) condition” should be corrected to be “due to ultra-high vacuum (UHV) condition”

--“The structure was growing and investigated for photo-detecting properties” should be corrected to be “The structure was grown and investigated for photo-detecting properties”

Author Response

Response to the Editor and Reviewers’ comments

Coatings

“Non-polar gallium nitride for photodetection applications: A systematic review”

Dear Editor and Reviewers,

Thank you for your useful comments and suggestion on the structure and scientific content of our manuscript. We have modified the manuscript accordingly, and detailed correction listed below point by point:

Reviewer #1:

Comments and Suggestions for Authors

In this work, studies about non-polar GaN based photodetector over a 7-year period were systematically reviewed. The reviewer found that this work will be of interest to other researchers in scientific and engineering community of III-V family compound semiconductors. In the view of my point, before the consideration for publication, the following issues needs to be addressed:

1) The introduction have room to be further improved. The authors write: “Over the past decade, III-V semiconductor materials have showed promising outcomes for the applications of optoelectronic devices, such as light-emitting diodes (LEDs), laser diodes (LDs) and photodetectors [1-7].” The general reference list seems a bit thin, considering the evolution in the field within the recent years. Several recent references concerning on GaN-based LEDs, such as Optics Express 27, A669 (2019); Applied Physics Letters 118, 182102 (2021); Nano Energy 2020, 69, 104427, etc., should be added, so that the readers can be clear about the state-of-the-art of this topic.

Action/amendment: Thank you for the highlighted comment. The comment has been addressed and several recent references have been included within the revised manuscript in page 1, introduction (third line) and highlighted in red.

2) The authors argued: “Different substrates were utilized based on the obtained articles, namely c-plane sapphire, r-plane sapphire, a-plane sapphire substrate and Si substrates as shown in Figure 3.” Can the authors include SiC substrate and GaN bulk substrate?

Action/amendment: Thank you for the highlighted comment. The comment has been addressed. The current study demonstrated the previous studies about non-polar GaN based photodetector over a 7-year period using the related academic digital databases were: i) Science direct, ii) Web of Science (WOS), iii) Institute of Physics (IoP), iv) American Institute of Physics (AIP), v) American Chemical Society (ACS) and vi) Wiley. Thus far, different substrates were covered based on the obtained articles, namely c-plane sapphire, r-plane sapphire, a-plane sapphire substrate and Si substrates. In addition, based on the obtained articles from the abovementioned academic digital databases, there were no studies about the use of SiC substrate and GaN bulk substrate. However, the use of SiC substrate and GaN bulk substrate has been highlighted in the introduction in page 2, line 33-36.

3) The author should provide schematic device structure of GaN-based UV detector in the manuscript. This will make this paper more readable.

Action/amendment: Thank you for the highlighted comment. The comment has been addressed and a schematics diagram of GaN-based photodetector has been included in page 7 as Figure 3.

4) What is a-GaN? Do the authors mean a-plane GaN?

Action/amendment: Thank you for the highlighted comment. The a-GaN stands for a-plane GaN. The comment has been addressed and the terms a-GaN have been changed to a-plane GaN to avoid any misunderstanding as highlighted within the revised manuscript.

5) The authors write: “An ultrahigh responsivity.” This sentence is incomplete.

Action/amendment: Thank you for the highlighted comment. We are sorry for the unintentional mistake. The comment has been addressed and corrected accordingly in page 19, line 27.

6) The authors wirte: “The growth of non-polar GaN on plana Si substrate” What is the meaning of plana Si substrate?

Action/amendment: Thank you for the highlighted comment. We are sorry for the unintentional mistake. The comment has been addressed and corrected accordingly to “the growth of non-polar GaN on Si substrate” as highlighted in page 35, line 7

7) “semiconductor substrates can be conducted due to their due to their optical and electronic properties” should be revised to be “semiconductor substrates can be conducted due to their optical and electronic properties”.

Action/amendment: Thank you for the highlighted comment. We are sorry for the unintentional mistake. The comment has been addressed and corrected accordingly to “semiconductor materials can be conducted due to their optical and electronic properties” within the revised manuscript in page 37, line 14-15.

8) “Hence, other types of metal should be further used the investigate the above phenomenon.” should be revised to be “Hence, other types of metal should be further used to investigate the above phenomenon.”

Action/amendment: Thank you for the highlighted comment. We are sorry for the unintentional mistake. The comment has been addressed and corrected accordingly to “Hence, other types of metal should be further used to investigate the above phenomenon” within the revised manuscript in page 37, line 39-40.

9) There are some grammatical errors in the manuscript, although most of them do not obscure the understanding of the technical contents. However, I believe that the paper should be proof-read for English before it is submitted. For example:

--“III-V semiconductor materials have showed promising…” should be corrected to be “III-V semiconductor materials have shown promising…”

Action/amendment: Thank you for the highlighted comment. We are sorry for the unintentional mistake. The comment has been addressed and corrected accordingly to “III-V semiconductor materials have shown promising” within the revised manuscript in page 1 (introduction, line 1)

--“The exciton binding energy of these materials are (40-52 meV)…” should be corrected to be “The exciton binding energies of these materials are (40-52 meV)…”

Action/amendment: Thank you for the highlighted comment. We are sorry for the unintentional mistake. The comment has been addressed and corrected accordingly to “The exciton binding energies of these materials are (40-52 meV) in page 2, line 3.

--“…impairing devices’ efficiency such response of rise and fall time…” should be corrected to be “…impairing devices’ efficiency such as response of rise and fall time…”

Action/amendment: Thank you for the highlighted comment. We are sorry for the unintentional mistake. The comment has been addressed and corrected accordingly to “impairing devices’ efficiency such as response of rise and fall time” within the revised manuscript in page 2, line 32.

--“…approach that can be utilize to determine a definitive topic in the literature…” should be corrected to be “…approach that can be utilized to determine a definitive topic in the literature…”

Action/amendment: Thank you for the highlighted comment. We are sorry for the unintentional mistake. The comment has been addressed and corrected accordingly to “approach that can be utilized to determine a definitive topic in the literature” within the revised manuscript in page 3, line 16.

--“…conducted on the on the five major databases within 6 years” should be corrected to be “…conducted on the five major databases within 6 years”

Action/amendment: Thank you for the highlighted comment. We are sorry for the unintentional mistake. The comment has been addressed and corrected accordingly to “conducted on the five major databases within 6 years” within the revised manuscript in page 3, line 37.

--“It should be noted that the Boolean operators used here owing to their data source and access capabilities” should be corrected to be “It should be noted that the Boolean operators were used here owing to their data source and access capabilities”

Action/amendment: Thank you for the highlighted comment. We are sorry for the unintentional mistake. The comment has been addressed and corrected accordingly to “It should be noted that the Boolean operators were used here owing to their data source and access capabilities” within the revised manuscript in page 3, last line.

--“More interestingly, the behavior of a-GaN epitaxial layer towards the extended defects were investigate” should be corrected to be “More interestingly, the behavior of a-GaN epitaxial layer towards the extended defects were investigated”

Action/amendment: Thank you for the highlighted comment. We are sorry for the unintentional mistake. The comment has been addressed and corrected accordingly to “More interestingly, the behavior of a-GaN epitaxial layer towards the extended defects were investigated” within the revised manuscript in page 9, line 12.

--“The work comprise non-polar a-plane GaN epitaxial films were grown on an r-plane sapphire” should be corrected to be “The work comprise non-polar a-plane GaN epitaxial films grown on an r-plane sapphire”

Action/amendment: Thank you for the highlighted comment. We are sorry for the unintentional mistake. The comment has been addressed and corrected accordingly to “The work comprise non-polar a-plane GaN epitaxial films grown on an r-plane sapphire” within the revised manuscript in page 9, line 37-38.

--“It is important to studies the effect…” should be corrected to be “It is important to study the effect…”

Action/amendment: Thank you for the highlighted comment. We are sorry for the unintentional mistake. The comment has been addressed and corrected accordingly to “Apart from that, the effect It is important to studies the effect of the dislo-cations and BSFs of the epitaxial layer grown by MOCVD towards the performance of the on performing of a-plane GaN-based a-GaN photodetectors in terms of its electrical properties was reported [30].” within the revised manuscript in page 14, line 43-46.

--“due to of ultra-high vacuum (UHV) condition” should be corrected to be “due to ultra-high vacuum (UHV) condition”

Action/amendment: Thank you for the highlighted comment. We are sorry for the unintentional mistake. The comment has been addressed and corrected accordingly to “due to ultra-high vacuum (UHV) condition” within the revised manuscript in page 17, line 23-24.

--“The structure was growing and investigated for photo-detecting properties” should be corrected to be “The structure was grown and investigated for photo-detecting properties”

Action/amendment: Thank you for the highlighted comment. We are sorry for the unintentional mistake. The comment has been addressed and corrected accordingly to “The structure was grown and investigated for photo-detecting properties” within the revised manuscript in page 11, line 29-30.

Finally, the English language of the whole manuscript has been edited.

Reviewer 2 Report

See file attached

Author Response

Response to the Editor and Reviewers’ comments

Coatings

“Non-polar gallium nitride for photodetection applications: A systematic review”

Dear Editor and Reviewers,

Thank you for your useful comments and suggestion on the structure and scientific content of our manuscript. We have modified the manuscript accordingly, and detailed correction listed below point by point:

Reviewer #2:

 Non-polar gallium nitride for photodetection applications: A systematic review

The authors, Al-Zuhari et al. present a review about non-polar gallium nitrite-based photodetectors. In this review the authors present information about the different synthesis process for these materials, as well as their structural and electrical properties. They also summarized the different challenges researchers have found at growing the material and making it into a device.

Broad Comments:

  1. It is assumed that the authors have done a comprehensive review of the literature, therefore section 2 is not needed in the manuscript. The journal is about coatings, so the main focus should be done in this direction. Additionally, the references in this section are missing some information; journal name or volume and pages.

Action/amendment: Thank you for the highlighted comment. The current study demonstrated the previous studies about non-polar GaN based photodetector over a 7-year period using the related academic digital databases were: i) Science direct, ii) Web of Science (WOS), iii) Institute of Physics (IoP), iv) American Institute of Physics (AIP), v) American Chemical Society (ACS) and vi) Wiley. Therefore, the aim was about a review of non-polar GaN-based photodetector by covering the above-mentioned academic databases, which is well correlated with the special issue of the Journal (Thin Films and Nanostructures by MOCVD: Fabrication, Characterization and Applications). Furthermore, the term comprehensive to replaced with systematically as highlighted in the abstract in page 1, line 9. In addition, the references of the section 2 have been corrected accordingly to include the journal name or volume and pages as highlighted in the list of reference.

  1. References need to be included in the first paragraph of section 3.1.

Action/amendment: Thank you for the highlighted comment. The comment has been addressed and reference in section 3.1 have been included in page 7, line 5, 9, 114.

  1. Check grammar and the flow of the paper, several sentences seem to be incomplete or are unclear. Here are few of them:

  1. Section 3.1.1. II: “In this work, 4 types of surface-engineered nanostructured along non-polar direction.” Something is missing here.

Action/amendment: Thank you for the highlighted comment. The comment has been addressed and the sentence has been corrected as highlighted in page 9, line 42-45.

  1. Section 3.1.2. “Apart from that, the use of a p-n junction geometry in NW’s design is the architecture with a p-n junction is perpendicular to the wire axis was helpful, in which it was not favourable since an additional leakage via surface states is present.”

Action/amendment: Thank you for the highlighted comment. The comment has been addressed and the sentence has been corrected as highlighted in page 12, line 1-4. The new sentence is “In contrast, the use of a p-n junction geometry in NW’s design, which was perpendicular to the wire axis was introduced. However, the structure was not favorable due to the presence of the additional leakage via surface states.

  1. “The combination of this category due to its easy fabrication and high flexibility of organic components with the superiority of electrical and optical properties of inorganic semiconductors”

Action/amendment: Thank you for the highlighted comment. The comment has been addressed and the sentence has been corrected as highlighted in page 12, line 31-34.

  1. Section 3.1.3. I: “In this work, the growth of high-quality non-polar a-plane GaN with a minimal set of pre-growth conditions using PAMBE.”

Action/amendment: Thank you for the highlighted comment. The comment has been addressed and the sentence has been corrected as highlighted in page 13, line 12-14.

Specific Comments:

  1. Line 17: flame detection is repeated.

Action/amendment: Thank you for the highlighted comment. The comment has been addressed and the repeated words were deleted as highlighted in the second line of the abstract.

  1. Lines 76-78: The sentence is not clear

Action/amendment: We are sorry, but we did not know where to find this sentence.

  1. Section 3.1.1. III first paragraph: which deposition technique, all of them? Or any in particular.

Action/amendment: Thank you for the highlighted comment. Sorry for this mistake. The comment has been addressed and the sentence was revised as highlighted in the second line of section 3.1.1. III in page 9.

  1. Section 3.1.1. IV, check the following sentence “… Si substrates by using our two-step-processes leads to a major improvement …”. I believe you are talking about someone else process not yours. Same issue in section 3.1.3. II.

Action/amendment: Thank you for the highlighted comment. Sorry for this mistake. The comment has been addressed and the sentence was revised as highlighted in section 3.1.1. IV (page 10, line 5) and 3.1.3. II. (page 14, line 5)

  1. Check the flow of the following sentence:

Finally, the English language of the whole manuscript has been edited.
